

**Archaeal Intact Polar Lipids in Polar Waters: A Comparison Between**
**the Amundsen and Scotia Seas**
Charlotte L. Spencer-Jones[1], Erin L. McClymont[1], Nicole J. Bale[2], Ellen C. Hopmans[2],
Stefan Schouten[2,3], Juliane Müller[4], E. Povl Abrahamsen[5], Claire Allen[5], Torsten
Bickert[4], Claus-Dieter Hillenbrand[5], Elaine Mawbey[5], Victoria Peck[5], Aleksandra
Svalova[6], James A. Smith[5]
[1]Department of Geography, Durham University, Lower Mountjoy, South Road, Durham, DH1 3LE, UK.
[2]NIOZ Royal Netherlands Institute for Sea Research, Department of Marine Microbiology and
Biogeochemistry, P.O. Box 59, 1790 AB Den Burg, Texel, The Netherlands.
[3] Department of Earth Sciences, Utrecht University, Utrecht, The Netherlands.
[4]Alfred Wegener Institute, Helmholtz Center for Polar and Marine Research, 27568 Bremerhaven, Germany.
[5]British Antarctic Survey, High Cross, Madingley Road, Cambridge, CB3 0ET, UK.
[6] School of Natural and Environmental Sciences, Newcastle University, Newcastle-upon-Tyne, NE1 7RU,
UK.
Correspondence to: Charlotte L. Spencer-Jones (charlotte.spencer-jones@open.ac.uk)
Abstract
The West Antarctic Ice Sheet (WAIS) is one of the largest potential sources of future sea-level rise, with
glaciers draining the WAIS thinning at an accelerating rate over the past 40 years. Due to difficulties in
calibrating palaeoceanographic proxies for the Southern Ocean, it remains difficult to assess whether similar
changes have occurred earlier during the Holocene or whether there is underlying centennial to millennial
scale forcing in oceanic variability. Archaeal lipid – based proxies, specifically Glycerol Dialkyl Glycerol
Tetraether (GDGT) (e.g. $TEX_{86}$ and $TEX_{86}^L$) are powerful tools for reconstructing ocean temperature, but
these proxies have been shown previously to be difficult to apply to the Southern Ocean. A greater
understanding of the parameters that control Southern Ocean GDGT distributions would improve the
application of these biomarker proxies and thus help provide a longer-term perspective on ocean forcing of
Antarctic ice sheet changes. In this study, we characterised intact polar lipid (IPL) - GDGTs, representing
(recently) living archaeal population in suspended particulate matter from the Amundsen Sea and the Scotia
Sea. Shifts in IPL-GDGT signatures across well-defined fronts of the Southern Ocean revealed a correlation
between the physicochemical parameters of these water masses and IPL-GDGT distributions. Further
analysis is required to elucidate the additional role of productivity and nutrient availability on Southern
Ocean IPL-GDGT distributions. Of particular note for proxy development in the Amundsen Sea is that IPL-





GDGTs are likely actively synthesised at Circumpolar Deep Water depths and may be a significant source of
GDGTs exported to the sedimentary record in this region.
Key words
Southern Ocean, Intact Polar Lipid (IPL), Glycerol Dialkyl Glycerol Tetraether (GDGT), Amundsen Sea,
Scotia Sea, Circumpolar Deep Water, Archaea, Thaumarchaeota.
**1.   Introduction**
Over the past ca. 50 years the West Antarctic Ice Sheet (WAIS) has lost ice mass at an accelerating rate with
some suggesting that the complete collapse of the WAIS may already be underway (Joughin et al., 2014;
Mouginot et al., 2014; Rignot et al., 2019). The WAIS is grounded below sea level and the edges of the ice
sheet are floating ice shelves that are, highly sensitive to changes in ocean properties. Widespread ice
sheet/shelf thinning will likely have influence on biogeochemical cycling through ocean productivity
(Raiswell et al., 2008; Menviel et al., 2010; Wadham et al., 2013), carbon reservoirs and carbon
sequestration (Yager et al., 2012; Wadham et al., 2019), in addition to sea ice and ocean circulation changes
(Menivel et al., 2010).
One of the challenges in understanding and predicting the behaviour of WAIS is a lack of long-term ocean
temperature records (i.e. prior to the satellite era ~1992). Such records are needed to better understand the
links between WAIS stability, physical properties of the Southern Ocean, and biogeochemistry which might
vary on centennial to millennial timescales (Smith et al., 2017; Hillenbrand et al., 2017). Organic
geochemical proxies based on the ratios of archaeal membrane lipids can be used to reconstruct past ocean
temperature and biogeochemistry. Glycerol dialkyl glycerol tetraether (GDGT) lipids are particularly
promising with the $TEX_{86}$, $TEX_{86}^{L}$ and OH-GDGT proxies having been widely used to reconstruct ocean
temperatures in tropical, temperate, and northern polar regions (e.g. Jenkyns et al., 2004; Huguet et al., 2006,
2011; Sinninghe Damsté et al., 2010; Darfeuil et al., 2016). In contrast, only a handful of studies have
successfully applied these proxies in the Southern Ocean (Kim et al., 2012; Shevenell et al., 2011; Etourneau
et al., 2013, 2019). This reflects a combination of low concentrations of GDGTs as well as an incomplete
understanding of archaeal populations and habitat/niche preference (Kim et al., 2010). A better
understanding of the source of GDGTs in the Southern Ocean and factors that impact archaeal populations
could improve application of $TEX_{86}$ based proxies in this environment.



## 1.1. Tracing Archaea with Intact Polar Lipids

Archaea are a key component of picoplankton within the polar oceans (Delong et al., 1994; Murray et al., 1998; Church et al., 2003; Kirchman et al., 2007; Alonso-Saez et al., 2008) and have an important role in biogeochemical cycling and in marine food webs. GDGTs are important cell membrane components present in many marine archaea (Schouten et al., 2013 and references therein) including the ammonia oxidising archaea (AOA) Thaumarchaeota (previously assigned to the phylum Crenarchaeota; Brochier-Armanet et al., 2008; Spang et al., 2010). Marine archaea produce isoprenoid GDGTs with a polar head group (intact polar lipids - IPLs). Upon cell death the polar head group is relatively rapidly cleaved off resulting in the preservation of the core GDGT lipid (c-GDGTs). c-GDGTs are subsequently preserved in the sedimentary record and can be used to reconstruct Antarctic palaeoenvironmental change over long time scales (Kim et al., 2012; Shevenell et al., 2011; Etourneau et al., 2013, 2019). Thaumarchaeota are a major source of GDGTs to the environment with pure culture studies detecting GDGTs with 0-3 cyclopentane moieties, crenarchaeol (cren, which contains 4 cyclopentane moieties and a cyclohexane moiety) and cren regio isomer (cren', Schouten et al., 2000; Sinninghe Damsté et al., 2018). Other archaeal phyla (e.g. marine Euryarchaeota group II) have been hypothesised as sources of GDGTs to the marine realm (Lincoln et al., 2014a,b), however this source is unlikely to be significant in marine samples (Schouten el. al., 2014; Zeng et al., 2019; Besseling et al., 2020). Furthermore, archaea exist throughout the marine water column with several studies suggesting a GDGT contribution to sediments from "deep water" Thaumarcheota (e.g. Ingalls et al., 2006; Shah et al., 2008; Kim et al., 2016).

IPL-GDGTs may be used as proxies for tracing (recently) living archaeal populations (e.g. Pitcher et al., 2011; Sinninghe Damsté et al., 2012; Elling et al., 2014, 2017). AOA enrichment cultures reveal three common GDGT head groups; monohexose (MH), dihexose (DH), and hexose-phosphohexose (HPH) (Schouten et al., 2008; Pitcher et al., 2010, 2011), with all three IPL head groups reported in environmental samples (Lipp et al., 2008; Lipp and Hinrichs, 2009; Schubotz et al., 2009; Schouten et al., 2012; Xie et al., 2014; Evans et al., 2017; Sollich et al., 2017; Besseling et al., 2018). HPHs are a common IPL in all AOA enrichment cultures, to date, with MH and DH intermittently present (Pitcher et al., 2011; Elling et al., 2017; Bale et al., 2019). The interpretation of IPL-GDGTs as proxies for living archaeal biomass is complicated by their degradation to c-GDGTs with increasing evidence that some IPLs are preserved following cell death



(Bauersachs et al., 2010; Huguet et al., 2010; Schouten et al., 2010; Xie et al., 2013; Lengger et al., 2014).
Kinetic modelling has suggested greater preservation of glycolipids compared with phospholipids (Schouten
et al., 2010), therefore suggesting that HPH-GDGTs may have potential as biomarkers for living,
metabolically active, Thaumarchaeotal populations (Schouten et al., 2012; Elling et al., 2014, 2017).
However, HPH-GDGT abundance is variable across the 1.1a Thaumarchaeota clade which could make the
interpretation of this biomarker in environmental studies complex (Elling et al., 2017). DH-GDGTs and DH-
OH-GDGT on the other hand are thought to be produced exclusively by 1.1a Thaumarchaeota with more
uniform abundance across the clade (Pitcher et al., 2011; Sinninghe Damsté et al., 2012), and could therefore
be potential tracers for living Thaumarchaeota (Elling et al., 2017).
In this study, we present the first characterisation of IPL-GDGTs in suspended particulate matter (SPM)
from two locations in the Southern Ocean, the Scotia Sea and the Amundsen Sea. The first aim of this study
is to characterise the distributions of IPL-GDGTs within the Southern Ocean in order to expand our
understanding of Thaumarchaeotal distributions in Polar Regions and improve our interpretation of GDGT
based proxies. The second aim of this study is to understand the environmental controls on IPL-GDGT
distributions in the Southern Ocean. In this study, we analyse the water column profiles of IPL-GDGTs with
18 samples from the Amundsen Sea and 15 samples from a transect in the Scotia and Weddell Sea.
**1.2. Study Area**
The Southern Ocean drives global thermohaline circulation and is therefore a major regulator of Earth's
oceans and climate (Carter et al., 2009). The clockwise flowing Antarctic Circumpolar Current (ACC)
connects all the major ocean basins resulting in a major role in the distribution of heat, salt, and gasses
(Carter et al., 2009). The surface waters of the Southern Ocean show clear shifts in water properties (salinity
and temperature) which mark ocean fronts, and in the present study include; Sub-Antarctic front (SAF), the
Polar Front (PF), the Southern Front of the ACC (SACCF), and the Southern Boundary of the ACC
(SBACC) (Carter et al., 2009 and references therein). Antarctic surface waters (AASW; 100m thick),
extending from the Antarctic continental shelf to the PF, are characterised by near freezing temperatures and
salinity values up to 34.3 practical salinity units (PSU), although these properties can vary on a regional basis
(Carter et al., 2009 and references therein). The transition between AASW south of the PF and Sub-Antarctic
surface water (SASW) north of the SAF occurs in the Polar Frontal Zone. Due to complex mixing processes,



the properties of surface water in the Polar Frontal Zone are often variable, but this water is generally
warmer (3-8 °C) and less dense (salinity 34-34.4 PSU) than AASW (Carter et al., 2009 and references
therein). Lastly, SASW is comparatively warmer (6-12 °C) with salinity >34.3 PSU (Carter et al., 2009 and
references therein). Circumpolar Deep Water (CDW) together with CDW-derived, modified deep-water
masses, such as Warm Deep Water in the Weddell Gyre (e.g. Vernet et al., 2019) is a key Southern Ocean
water mass and can be detected between ~1400 m and >3500 m depth offshore from the Antarctic continent.
CDW can rise to meet AASW or even outcrop along the Antarctic continental margin (Carter et al., 2009 and
references therein). Mixing of CDW with different water masses gives rise to two types: Upper CDW
(UCDW) defined by an oxygen minimum, high nutrient concentrations, and a depth of 1400-2500 m; and
Lower CDW (LCDW) defined by a salinity maximum of 34.70-34.75 PSU (Carter et al., 2009 and
references therein). In contrast to UCDW, LCDW extends south of the SBACC (Orsi et al., 1995), is
upwelling at the continental slope, and can protrude onto the shelf where it mixes with super cooled shelf
waters, renewing LCDW and forming Antarctic Bottom Water (AABW) (Carter et al., 2009 and references
therein).
The Scotia Sea is located in the eastern Atlantic sector of the Southern Ocean (20°W to 65°W) bounded by
the South Atlantic Ocean to the North, the Drake Passage to the West, and by the Weddell Sea to the South
(Figure 1). The Scotia Sea is influenced by the eastward flow of the ACC, via the Drake Passage, and by a
northward component of the ACC, caused by topographic steering and northward outflow of recently
ventilated waters from the Weddell Sea , whereby Weddell Sea Deep Water (WSDW) is incorporated into
the ACC (Locarnini et al., 1993; Naveira Grabato et al., 2002a,b), thus creating a region of high mixing
(Heywood et al., 2002) and intense water mass modification (Locarnini et al., 1993).
The Amundsen Sea extends from 100°W to 130°W and is bounded by the Sub-Antarctic Pacific to the North
(Figure 1). The Amundsen Sea water column south of the PF mainly consist of a thin upper layer of cold and
fresh AASW overlying relatively warm CDW. The Amundsen Sea embayment is located offshore from one
of the major WAIS drainage basins and observations show a clear trend in glacial retreat over recent decades
(e.g. Mouginot et al., 2014; Paolo et al., 2015; Rignot et al., 2019). The deep ice shelves (extending up to
1000 m below sea level) surrounding the Amundsen Sea embayment are exposed to unmodified CDW which
can be up to 4 °C above the *in situ* melting point (Jacobs et al., 1996, 2011; Rignot and Jacobs, 2002; Jenkins





et al., 2010; Rignot et al., 2013; Webber et al., 2017) so that CDW may drive enhanced melt rates and ice
sheet instability in this region (Shepherd et al., 2001; Zwally et al., 2005; Rignot et al., 2008; Pritchard et al.,
2009; Wingham et al., 2009).
**2.  Methodology**
**2.1. Sample collection**
A Seabird Scientific SBE911plus conductivity-temperature-depth (CTD) instrument with a 24 bottle rosette
was used to vertically profile the water column and collect water for organic geochemical analysis. Water
was collected on board the *RRS* James Clark Ross (expeditions JR272 and JR257) during March-April 2012
(austral autumn) from 15 stations along the former WOCE A23 section (Meredith et al., 2001) traversing the
Scotia Sea between the northern Weddell Sea and South Georgia (Table 1 and Figure 1; Allen et al., 2012;
Venables et al., 2012), and on board the *R/V* Polarstern expedition PS104 during February-March 2017
(austral summer) from 5 stations in the Amundsen Sea embayment (Table 2 and Figure 1; Gohl, 2017).
Water samples were collected in 10 L Niskin bottles. In the Scotia Sea, the depth of the sample collection
was dependent on the expression of the mixed layer and seasonal thermocline as observed during each CTD
deployment. At all stations, a "mixed layer" sample was collected between 10-40m depth and a "thermocline
layer" sample collected between approximately 60-110 m depth (Table 1). In the Amundsen Sea, the
sampling strategy included samples from surface thermocline waters, and CDW. Water samples
(approximately 10-30 L) were vacuum filtered through pre-combusted GF/F filters (Whatman, 0.7 µm pore
size, 50 mm diameter). Glass fibre filters with a nominal pore size of 0.7 µm are most commonly used for
sampling of SPM in ocean and lake waters. However, as microbes can range in size from 0.2-0.7 µm, these
filters may lead to an under-sampling of archaeal cells that are not associated with aggregates (Lee et al.,
1995; Ingalls et al., 2012). Therefore, IPL-GDGT concentrations reported here represent the minimum likely
concentrations.
The filters were subsequently stored in foil at -20 °C, then transported to Durham University (UK; Scotia sea
samples) and Alfred Wegener Institute (Germany; Amundsen Sea samples). Samples were freeze-dried prior
to lipid extraction.



### 2.2. Sample extraction

Total lipids of the Scotia Sea sample set were extracted at the Royal Netherlands Institute for Sea Research.
Freeze-dried samples were extracted using a modified Bligh and Dyer methodology as detailed in Besseling
et al. (2018). Briefly, sample filters were cut into small pieces using solvent cleaned scissors. The total lipids
were extracted using a monophasic mixture of $K_2HPO_4$ (8 g/L adjusted to pH 7-8), dichloromethane
($CH_2Cl_2$) and methanol ($CH_3OH$) at a ratio of 0.8:1:2. Extractions were repeated three times and pooled. The
pooled extract was subsequently phase separated by adjusting the ratio of $K_2HPO_4$: $CH_2Cl_2$: $CH_3OH$ to
0.9:1:1. The $CH_2Cl_2$ layer of the resultant bi-phasic mixture was transferred to a round bottom flask. This
was repeated three times, with the Bligh Dyer extract (BDE) dried under a stream of $N_2$.
Total lipids of the Amundsen Sea sample set were extracted at the Alfred Wegener Institute (Germany).
Freeze dried samples were extracted ultrasonically using $CH_2Cl_2$ and $CH_3OH$ at a ratio of 2:1 for 15 minutes.
This was repeated three times, the extracts pooled and dried under a stream of $N_2$. The resulting total lipid
extract was fractionated over a silica column using hexane (for elution of the alkanes and highly branched
isoprenoids) followed by $CH_2Cl_2$:hexane and $CH_2Cl_2$:$CH_3OH$ both at a ratio of 1:1 for elution of the polar
fraction. The polar fraction was dried under $N_2$ and stored at -20 °C prior to IPL analysis. The method used
for the extraction of the Amundsen Sea samples is not the regular IPL extraction procedure as it, for
example, does not use phosphate buffer and that this may have an influence of the results obtained for the
Amundsen Sea samples.
An internal standard of 1-O-hexadecyl-2-acetyl-*sn*-glycero-3-phosphocholine was added to both the
Amundsen and Scotia Sea samples. The TLE (Scotia Sea) and polar fraction (Amundsen Sea) were filtered
through true regenerated cellulose filters (4 mm, 0.45 μm pore size) using hexane, propan-2-ol, and water at
a ratio of 79:20:1. Samples were stored at -20 °C prior to analysis.

### 2.3. Intact Polar Lipid characterisation

IPL-GDGTs were analysed using a modification of the Sturt et al. (2004) methodology as detailed in
Besseling et al. (2018). To summarise, an Agilent 1290 Infinity I UHPLC, equipped with a thermostated
auto-injector and column oven, coupled to a Q Exactive Orbitrap MS with Ion Max source with a heated
electrospray ionisation (HESI) probe (Thermo Fisher Scientific, Waltham, MA, USA). Separation was
achieved using a YMC-Triart Diol-HILIC column (250 x 2.0 mm, 1.9 μm particle size, 12 nm pore size;



YMC co., Ltd., Kyoto, Japan) maintained at 30 °C with a flow rate of 0.2 mL/min. Chromatographic
separation of IPL-GDGTs was achieved using the following 70 minute program: 0% eluent B from 0-5
minutes, linear gradient to 34% eluent B at 25 minutes, isocratic 25-40 minutes, linear gradient to 60% B at
55 minutes, linear gradient to 70% B 65 minutes, followed by a re-equilibration time of 20 minutes between
each analysis. Eluent A was hexane/propan-2-ol/formic acid/ 14.8 M $NH_{3aq}$ (79:20:0.12:0.04 [v/v/v/v]),
eluent B is propan-2-ol/water/formic acid/14.8 M $NH_{3aq}$ (88:10:0.12:0.04 [v/v/v/v]). HESI sheath gas,
auxiliary gas and sweep gas $N_2$ pressures were 35, 10, and 10 (arbitrary units) respectively with the auxiliary
gas at 50 °C. The spray voltage was 4.0 kV (positive ion ESI), S-Lens 70 V, and capillary temperature 275
°C. Mass range monitored was between *m/z* 375 and 2000 (resolving power of 70 000 ppm at *m/z* 200)
followed by data dependent fragmentation of the 10 most abundant masses in the mass spectrum (with the
exclusion of isotope peaks) were fragmented successively (stepped normalised collision energy 15, 22.5, 30;
isolation window 1.0 m/z). A dynamic exclusion window of 6 s was used as well as an inclusion list with a
mass tolerance of 3 ppm to target specific compounds (absolute *m/z* values of IPL-GDGTs can be found in
supplement A). The Q Exactive Orbitrap MS was calibrated within a mass accuracy range of 1 ppm using the
Thermo Scientific Pierce LTQ Velos ESI Positive Ion Calibration Solution (containing a mixture of caffeine,
MRFA, Ultramark 1621, and N-butylamine in an acetonitrile-methanol-acetic acid solution). Peak areas for
each individual IPL were determined by integrating the combined mass chromatograms (within 3 ppm) of
the monoisotopic and first isotope peak of all the relevant adducts formed (protonated, ammoniated, and/or
sodiated).  IPL-GDGTs were examined in terms of their MS peak area response. Thus, the relative
abundance of the peak area does not necessarily reflect the actual relative abundance of the different IPL-
GDGTs, however, this method allows for the comparison between samples analysed in this study. The peak
areas were determined from extracted ion chromatograms of the $[M+H]^+$, $[M+NH_4]^+$, and $[M+Na]^+$ for each
individual IPL-GDGT species. C-GDGT lipids were not analysed.
**2.4. Data Analysis**
Standards for individual IPL-GDGTs are not available therefore; concentrations reported here are semi-
quantitative. IPL-GDGT peak areas were normalised to the internal standard and volume of water filtered
and are reported as units/L. The Ring Index (RI) was calculated based on Zhang et al. (2016).



Redundancy analysis (RDA) was performed on the Scotia Sea data set in RStudio (version 1.2.1335) using
Vegan and Faraway packages. RDA was performed using data normalised to the internal standard and total
water volume extracted (scaled). Temperature, salinity, oxygen concentration, and Chlorophyll A
fluorescence (hereafter referred to as fluorescence) were selected as explanatory variables and IPL-GDGT
relative abundances are the response variables. Statistical significance of RDA, axes, and explanatory
variables were determined using an Anova-like test (Legendre et al., 2011).
**3.  Results**
**3.1. Physicochemical properties of the water column**
CTD measurements were taken at all 5 stations in the Amundsen Sea; PS104/003, PS104/007, PS104/017,
PS104/022, PS104/043. Temperature – salinity (T-S) plots are shown in Figure 2. Temperature, salinity,
fluorescence, oxygen profiles are shown for each of the Amundsen Sea stations in Supplement B S1. At the
time of sampling, water masses in the Amundsen Sea study area were characterised by a temperature range
of -1.7 to +1.1 °C, a salinity range of  32.8 to 34.7 PSU, and an oxygen concentration of between 4.5 and 8.9
ml/L (Supplement B S1). Ocean temperatures in the Amundsen Sea, generally show a relatively warm
surface layer, followed by a steep thermocline to a temperature minimum (winter water) extending down to
approximately 400m, with ocean temperatures increasing below this point due to the intrusion of relatively
warm CDW (Supplement B S1). Three different water masses are detected in the Amundsen Sea from the T-
S plot: AASW, CDW, and modified CDW (Figure 2). Fluorescence peaked at the surface within the
uppermost 20 m, followed by a steep decline with depth (Supplement B S1). High fluorescence values were
observed at PS104/017 with 8mg/m$^3$, and PS104/007 with 4 mg/m$^3$ respectively, whereas low fluorescence
values were observed at stations PS104/003, PS104/022, and PS104/043 (Supplement B S1).
The Scotia Sea study area encompasses the SAF, PF, SACCF and the SBACC (Figure 1a) and is
characterised by a temperature range of -1.6 to +7.3 °C, and a salinity range of 33.6-34.7 PSU (Figure 2).
The temperature range of the mixed layer samples was -1.2 to +7.3 °C and thermocline samples was -1.6 to
+6.1 °C. A clear partition between the sample stations is observed in the T-S plot (Figure 2) with consistently
higher water temperatures found at stations north of CTD 19 and on average lower ocean temperatures south
of CTD 18. This region broadly marks the location of the SBACC at ~58.6 °S (Figure 1a).



### 3.2. Intact Polar GDGT inventory

A total of 12 IPL-GDGTs (see Supplement B S2 for structures) were identified within 48 samples from the Southern Ocean (Figure 3 and 4, Tables 3 and 4). This included IPLs with cores comprising of GDGT-0, GDGT-1, GDGT-2, cren, OH-GDGTs-0, and diOH-GDGT-0. It should be noted that the LC-MS method utilised in this study does not separate individual GDGT isomers, including crenarchaeol (cren) and its isomer, cren', and hence the cren detected here is likely a combination of both isomers. The majority of the IPL-GDGTs in both the Amundsen and Scotia seas were regular GDGTs (i.e. not hydroxylated) with a mean contribution of 69% (±20%) and 66% (±18%) respectively (excluding samples where IPL-GDGTs were absent). diOHGDGT-0 was the least common core lipid identified in the Scotia and Amundsen Sea. Both forms of hydroxyl-GDGT were observed with zero cyclopentane moieties (OH-GDGT-0 and diOH- GDGT-0). While GDGT-1, GDGT-2, and cren were detected, GDGT-0 was the dominant regular GDGT in both the Amundsen and Scotia seas. GDGT-0, cren, and OH-GDGT-0 were all detected in combination with the MH, DH, and HPH head groups. GDGT-1 and GDGT-2 were only found to be associated with DH and diOH-GDGT-0 was only found in combination with MH (Tables 2 and 3).

### 3.3. Amundsen Sea depth profiles

Archaeal IPLs were identified in the water column at all Amundsen Sea stations. The relative abundance of the regular GDGT core (i.e. non-hydroxylated) varied with depth ranging from 20-100% of total IPL-GDGTs (excluding depths where no IPL-GDGTs were identified; Table 4). PS104/003 and PS104/007 were found to have IPL-GDGTs in the uppermost surface sample (10 m and 20 m depths respectively). The surface sample at PS104/003 (10m) was dominated by non-hydroxylated GDGTs (94.3% of total IPLs) with a lower relative abundance of OH-GDGT core type (5.7% of total IPLs). Further to this, HPH-GDGT-0 was the most abundant IPL-GDGT at this station (81.8% of total IPLs) with HPH-cren contributing a smaller fraction of the total IPL-GDGTs (11.1%). Low relative abundance of MH-GDGT-0 (<1%), MH-cren (<1%), MH-OH-GDGT-0 (<1%), DH-OH-GDGT-0 (5.1%), and MH-diOH-GDGT-0 (<1%) were also observed at PS104/003 10 m. This contrasts with the surface sample at PS104/007 (20 m) where no OH-GDGT-IPLs were detected and where the IPL-GDGT suite is split between MH-GDGT-0 (89.1%) and MH-cren (10.9%). IPL-GDGTs were not identified within the surface sample at PS104/017 (10 m) and the two mid-shelf stations, PS104/022 (10 m and 30 m) and PS104/043 (10 m). DH-GDGT-0 and DH-cren are minor



components of the IPL-GDGT suite with maximum relative abundance observed in the deepest samples for
all Amundsen Sea stations. The relative abundance of IPL-GDGTs with a MH head group peaks in the mid
depths between 120 and 240 m (with the exception of the surface 20 m at PS104/007). Further to this, the
ratio of MH/DH and MH/HPH is also highest at the mid depths between 120 and 240 m (Table 5 and 6). The
ratio of GDGT-0/cren is variable throughout the Amundsen Sea stations, ranging from 2.8-8.2 (excluding
samples with no GDGTs). The sample taken from 180 m water depth at PS104/003 exceeded this range with
a GDGT-0/cren ratio of 27.0 (Table 5 and 6).
**3.4. Scotia Sea transect**
Archaeal IPLs were detected within all 16 Scotia Sea stations. A clear depth trend in IPL-GDGTs can be
observed where IPL-GDGTs were detected in the thermocline samples but were often below detection within
the mixed layer. Exceptions to this are CTD 1, 16, 20, and 21 where IPL-GDGTs were present in both the
mixed and thermocline layers. Relative abundance (%) of IPL-GDGT cores and the degree of cyclicity
remains constant along the Scotia Sea transect with IPL-GDGT head groups showing greater variation along
the transect. An increase in the relative abundance of the HPH head group is observed within the thermocline
samples between CTD 22 (53.5 °S) and 5 (63.3 °S) this is coupled with a decrease in the relative abundance
(%) of MH and DH IPL-GDGT head groups (Figure 3b). Mixed layer CTD 20 and 21 are dominated by MH,
CTD 16 is dominated by HPH, and CTD 1 mixed layer contains a mixture of all three IPL-GDGT head
groups. The GDGT-0/cren ratio generally ranges from 1.6-9.9, but CTD 7 (21.7), 10 (177.6), and 16 (16.8),
located at the thermocline, exceed this range due to low cren concentrations (Table 5). In preparation for
RDA, biomarkers that were identified in fewer than three samples were designated "rare species" and were
excluded from the analysis (GDGT-DH-0, GDGT-DH-1 and OH-GDGT-HPH-0 excluded). This is because
outliers can violate the linearity of the relationship between the response and explanatory variables
(Legendre & Legendre, 2012). Samples 1 and 25 were also excluded from the analysis. Sample 1 is located
offshore of the Falkland Islands and is the only sample from North of the SAF, thus representing the only
data point for the Subantarctic Zone of the Southern Ocean that is unlikely to be representative for the polar
environment. Sample 25, located close to South Georgia, was excluded due to high biomarker abundances
(Figure 3a) which could be due to exceptionally high productivity in this area (e.g. Atkinson et al., 2001).
Variance inflation factors (VIFs) for the response variables were between 3.5 (fluorescence) and 11.4



(oxygen concentration) (Supplement C). The VIF for oxygen concentration is slightly higher than is typically
acceptable for RDA analysis. This is due to correlation between oxygen concentration and fluorescence
($R_2$=0.62), however, as the $R_2$ is below 0.7 this is unlikely to violate the assumptions of the RDA (Legendre
& Legendre, 2012). RDA shows 64% constrained variation with RDA1 and 2 accounting for 63% of the
cumulative variation. The RDA is statistically significant (p=<0.05, f=3.5), furthermore, RDA1 is found to
be statistically significant (p=<0.05, f=11.48) however, RDA2 is not significant (p=0.42, f=2.35). Species
scores show HPH-GDGT-0 and HPH-cren to load positive on RDA 1, with MH-GDGT-0, MH-cren, MH-
OH-GDGT-MH-0, DH-OH-GDGT-0, and MH-MH-diOH-GDGT-0 loading highly negative on RDA1. Of
the explanatory variables tested, temperature is statistically significant at the <0.05 level (f=8.56) and with
salinity (p=0.07, f=2.61) and oxygen concentration (p=0.09, f=2.58) approaching significance. The site
scores show CTD 20, 21, 22, 23, and 24 to be negatively loaded on RDA1 with CTD 3, 5, 7, 10, 13, 16, 18
and 19 to be positively loaded on RDA1 suggesting that these stations are contrasted along this axis.
**4.  Discussion**
**4.1. Hydroxylated GDGTs in Polar Environments**
In this study, two hydroxylated GDGTs (OH-GDGT-0 and diOH-GDGT-0) were detected. Hydroxylated
GDGTs have been reported as potential biomarkers for reconstructing ocean temperature change in cold
waters (Fietz et al., 2013, 2016) and in this study contribute up to 49.8% (OH-GDGT) and 30.1% (diOH-
GDGT) of total IPL-GDGTs. Hydroxylated IPL-GDGTs are not commonly reported in previous SPM
studies (e.g. Kim et al., 2016; Kang et al., 2017; Hurley et al., 2018). However, these compounds have been
reported as c-GDGTs in marine and lacustrine sediments, with hydroxylated GDGTs found to contribute
approximately 8% in marine sediments from temperate and tropical sites (Liu et al., 2012; Lu et al., 2015).
These compounds have been reported in much higher abundance in polar environments including up to 20%
in SPM and up to 16% in surface sediments from the Nordic Seas (Fietz et al., 2013) and up to 20% in
surface sediments from the Southern Ocean (Huguet et al., 2013).
In the Amundsen and Scotia seas, hydroxylated GDGTs made up a significant amount of the total IPL-
GDGT profile, contributing up to 79.9% (Amundsen Sea, PS104/003 - 180 m). Even excluding this one
exceptional sample, hydroxylated IPL-GDGT abundances were still consistently above 20% with a
maximum relative abundance of 48.1% in the Amundsen Sea and 68.5% of total IPL-GDGTs in the Scotia





Sea (Table 3 and 4). Exceptionally high hydroxylated GDGT relative abundances could be due to differences
in methodologies to the previous studies which measured core GDGTs by atmospheric pressure chemical
ionisation (APCI; Liu et al., 2012; Fietz et al., 2013; Huguet et al., 2013; Lu et al., 2015) while this study
examined IPL-GDGTs using electrospray ionisation (ESI). Using the same LC-MS methodology, Sollai et
al. (2019a) report average hydroxylated IPL-GDGT relative abundances of 22% (±19%) with a range of 0-
51% in SPM from the euxinic Black Sea; however, similar analyses from the Arabian Sea (Besseling et al.,
2018), the eastern tropical South Pacific (Sollai et al., 2019b) and the Mediterranean Sea did not detect
hydroxylated IPL-GDGTs. Molecular dynamics simulations have shown that the addition of hydroxyl
moieties in the tetraether structure increases the fluidity of the cell membrane and aid trans-membrane
transport in cold environments (Huguet et al., 2017). The exceptionally high amount of hydroxylated IPL-
GDGT for the Amundsen and Scotia seas may therefore be due to elevated synthesis of these biomarkers in
cold environments.
**4.2. IPL-GDGT Distributions as an Indicator of Archaeal Populations**
In both the Amundsen and Scotia Sea samples GDGT-0 dominated the IPL-GDGT profiles contributing up
to 89.1% of the total in the Amundsen Sea and up to 100% in the Scotia Sea. Low diversity of cyclic GDGTs
in this study is indicated by RI ranging from 0.02 – 1 for the Scotia Sea and 0.03 – 0.9 for the Amundsen Sea
(Tables 5 and 6). This is particularly low compared with the RI of the global core top calibration, which
includes a range of Southern Ocean samples, reporting an RI range of 1.25-3 (excluding the Red Sea
samples) (Kim et al., 2010; Ho et al., 2011, 2014; Zhang et al., 2016). Previous SPM studies spanning a
range of marine habitats have reported the presence of hydroxylated GDGT-1, -2, and -3 as well as a wider
range of non-hydroxylated GDGTs, such as GDGT-3 and -4 (Kim et al., 2016; Besseling et al., 2018; Hurley
et al., 2018; Sollai et al., 2019a,b). As this study used the same analytical methodology as Besseling et al.
(2018) and Sollai et al. (2019a,b), these differences cannot be attributed to analytical methodologies. Low
cyclic diversity of GDGTs in the Amundsen and Scotia seas could be due to differences in the synthesis of
these lipids by the source Thaumarchaeota. The relationship between ocean temperature and the cyclicity of
GDGTs has been firmly established, with increasing ocean temperatures correlated with increasing relative
abundance of GDGTs with 2 or more cyclopentane moieties (Schouten et al., 2002, 2007; Kim et al., 2008,
2010). However, Kim et al. (2010) note some differences between sub-tropical and sub-polar oceans, with





cren playing a more important role in temperature reconstructions in the subtropics than in polar oceans,
suggesting that there may be differences in membrane adaptation strategies of Thaumarchaeota. Principal
component analysis of IPL-GDGT distributions of a moderately thermophilic Thaumarchaeota along with
previously published data identifies two distinct clusters with a clear partition between the orders of
Nitrosopumilales and Nitrososphaeales (Bale et al., 2019). GDGTs analysed in this study cluster within the
Nitrosopumilales group due to the high relative abundances of GDGT-0 and low relative abundances of all
other GDGTs. Due to the polar locations of the Amundsen and Scotia Sea samples, Nitrosopumilales are
likely to be the key AOA in these environments. Previous microbial analysis of the spatial variation in
prokaryotes of the Amundsen Sea polynya identified the most abundant Thaumarchaea marine group I
(MGI) sequence belonged to the cluster affiliated with "*Ca. Nitrosopumilus maritimus*" (Kim et al., 2014). In
similar studies within the wider Southern Ocean region phylogenetic analysis reveals high abundances of
sequences clustering with *Nitrosopumilus*. Hernandez et al. (2015) analysed surface water samples from
Potter Cove (King George Island, wester Antarctica Peninsula) which revealed that the majority of sequences
fell into the clade containing "Ca. *Nitrosopumilus maritimus"* and other environmental sequences containing
Thaumarchaeota. Signori et al. (2018) studied microbial spatial and temporal variability at 10 stations off the
Antarctic peninsula revealing spring to be characterised by SAR11 and microbial communities remaining
from winter, including Thaumarchaeota (*Nitrosopumilus*), Euryarchaeota, and SAR324, with a shift in
microbial populations during the summer and autumn.
Three polar head groups were detected in this study, i.e. MH, DH, and HPH. All three head groups have
previously been identified in culture (Schouten et al., 2008; Pitcher et al., 2011; Sinninghe Damsté et al.,
2012; Elling et al., 2017), environmental studies (e.g. Zhu et al., 2016; Besseling et al., 2018), and have
widely been associated with Thaumarchaeota. It has been postulated that specific IPL-GDGTs may be
associated with particular Thaumarchaeotal groups or habitats (Sinninghe Damsté et al., 2012; Elling et al.,
2017; Bale et al., 2019). Previously the HPH head group has been associated with the Nitrosopumilales order
(Group I.1a) and the DH head group with the Nitrosophaeales order (Group I.1b) (Sinninghe Damsté et al.,
2012). More recent studies have shown that environmental niche or habitat may be the main driver of GDGT
head group composition rather than phylogeny (Elling et al., 2017; Bale et al., 2019). Relevant to this study,
Elling et al. (2017) analysed the lipidome of 10 Thaumarcheotal cultures and identified DH-GDGTs and DH-
OH-GDGTs as key membrane components of the marine mesophiles compared with the terrestrial



thermophilic and soil mesophilic Thaumarchaeota. In the present study, high abundances of HPH were
detected, contributing up to 92.9% and up to 100% of total IPL-GDGTs in the Amundsen Sea and Scotia Sea
respectively. The dominance of HPH in the lipid profiles of the Amundsen and Scotia seas align with
previous culture analysis (Schouten et al., 2008; Pitcher et al., 2011; Sinninghe Damsté et al., 2012; Elling et
al., 2017).

**4.3. Influence of Circumpolar Deep Water on IPL Distributions: Amundsen Sea**

In this study, we observed a number of consistent trends in the water column IPL-GDGT distributions
between the different Amundsen Sea sampling stations. In the surface samples, collected within the euphotic
zone at PS104/017 (10 m), PS104/022 (10 m and 30 m), PS104/043(10 m), no IPL-GDGTs were identified.
Previous studies from the Southern Ocean have shown water column archaeal distributions to be highly
variable on both a temporal and spatial scale. Broadly, archaea (as measured by cell counts or rRNA) are
often absent or found in relatively low abundance in the surface waters during the austral spring algal bloom
and during austral summer (Massana et al., 1998; Church et al., 2003; Kalanetra et al., 2009; Besseling et al.,
2020). The absence of archaea in the surface waters of the Southern Ocean contrasts with the high abundance
of bacteria and is part of a larger seasonal cycle in archaeal population dynamics (Church et al., 2003).
Temporal distributions of archaea are then shown to become more evenly distributed by depth, with an
increase in the population within the surface waters throughout austral autumn-winter (Church et al., 2003).
The Amundsen Sea samples were collected during austral summer. Two previous studies in the Antarctic
Peninsula show an increase in group I archaeal populations in surface waters during austral summer and
winter (Massana et al., 1998; Murray et al., 1998). However, Kalanetra et al. (2009) did not observe any
archaea in surface waters west of the Antarctic Peninsula during austral summer. The mechanism for this
temporal heterogeneity is likely mediated by a combination of physical and biological factors including,
water mass properties, concentrations of dissolved and particulate organic carbon (Murray et al., 1998).
Furthermore, the absence of AOA in the surface waters during austral spring, when primary productivity is
highest, could be due to competition with bacteria and algae that bloom during the same time period and/or a
subsequent nutrient limitation (Massana et al., 1998; Church et al., 2003; Kalanetra et al., 2009). As the
current study was only performed at one time point during austral summer a larger sampling campaign
would be required to fully characterise microbial and IPL-GDGT seasonality in the Amundsen Sea.



In contrast to the other stations, the surface water samples from PS104/003 and PS104/007 (10 m and 20 m
respectively) were found to contain IPLs. Unusually, the samples from PS104/007 (10 m) only contained the
MH head group. It should be noted that while the MH head group is known to be synthesised by archaea
(e.g. Sinninghe Damsté et al., 2012), this IPL is recalcitrant and can be formed as a degradation product of
other IPL-GDGTs (e.g. Lengger et al., 2013, 2014). In contrast, HPH is more labile and less readily
preserved in sediments following cell death and is hence considered to be a biomarker for recently active
archaea and, in particular, Thaumarchaeota (Pitcher et al., 2010; Sinninghe Damsté et al., 2012). While DH
head group is not as labile as HPH due to its glycosidic structure (Lengger et al., 2013), the prevalence of
DH across the Thaumarchaeota phylum may suggest some use of this head group as a biomarker for the
archaeal community (Elling et al., 2017). Hence, the dominance of the MH head group at this station may
indicate an inactive/relic archaeal population at this depth. Higher IPL-GDGT diversity was detected at
PS104/003 including HPH and DH head groups indicating a recently active archaeal population (Sinninghe
Damsté et al., 2012; Elling et al., 2017). PS104/003 is located in an area of active upwelling of nutrient-rich
waters largely composed of CDW (Pine Island Bay polynya) (Mankoff et al., 2012). Together with the
Amundsen Polynya located north of Dotson and westernmost Getz ice shelves (Figure 1), it is one of the
most productive regions (per unit area) of the Southern Ocean (Arrigo and van Dijken, 2003). Productivity is
further aided by the influx of iron released from the rapidly melting Thwaites and Pine Island glaciers
(Alderkamp et al., 2012; Gerringa et al., 2012; Thuroczy et al., 2012; St-Laurent et al., 2017). Productivity in
this area is not only limited by nutrient and iron availability but also by light; productivity is 30-50% lower
in the Pine Island Polynya compared to the Amundsen Polynya, with this difference attributed to the
significant difference in solar irradiance levels between the two polynyas throughout the summer season
(Park et al., 2017). Kalanetra et al. (2009) suggests that a combination of both light and nutrient differences
between Arctic and Antarctic ocean settings could cause the differences in archaeal populations in the
surface ocean, where low light and nutrient levels in the surface allows archaeal populations to flourish, with
further studies suggesting photoinhibition of Thaumarchaeota (Church et al., 2003; Mincer et al., 2007; Hu et
al., 2011; Merbt et al., 2012; Luo et al., 2014).
IPL-GDGT diversity increased downwards in the water column through the thermocline and the CDW layer
in the Amundsen Sea (Figure 4). DH-cren and HPH-cren may be widely applied as biomarkers for recently
active Thaumarchaeota populations having been identified as key cell membrane lipids (Pitcher et al., 2010;



Sinninghe Damsté et al., 2012; Elling et al., 2017). HPH-cren was identified consistently throughout the
thermocline and CDW layer at all Amundsen Sea stations (Table 4). Our results, therefore, suggest recently
active AOA at the thermocline and within the CDW. Tolar et al. (2016) shows ammonia oxidation (AO) to
occur throughout the water column, with similar rates of AO in CDW during both winter and summer
seasons and increased AO in surface waters during the late winter in sites west of the Antarctic Peninsula.
This is consistent with molecular microbiology studies from the Amundsen Sea and Antarctic Peninsula
region that identify Thaumarchaeota throughout the water column, but with a seasonal trend where these
archaea are often absent in the surface waters during spring and summer, and present in the CDW throughout
the season (Massana et al., 1998; Alonso-Saez et al., 2011). HPH-cren, however, may not be the most
suitable proxy for tracking the complete AOA population as the relative abundance of this IPL can vary
significantly between phylogenetic subgroups (Elling et al., 2017). DH-GDGTs have been identified with
consistent relative abundances across the Nitrosopumilales order (Group 1.1a), suggesting DH-cren as an
additional biomarker for AOA activity (Elling et al., 2017). In this study we detect DH-cren consistently in
the CDW layer and with low relative abundance in the thermocline of PS104/003 and PS104/007 and
absence in the thermocline waters at PS104/017 and PS104/022. Thaumarchaeota are thought to partition
between shallow water (0-130 m) and deep water (500-4000 m) marine clades (Francis et al., 2005; Hallam
et al., 2006). Therefore, this difference in HPH-cren and DH-cren distributions could reflect differences in
Thaumarchaeota populations in the Amundsen Sea. While the data presented here provide only a snapshot of
the Amundsen Sea IPL-GDGT distributions, this small contrast in HPH and DH-cren distributions may
represent a significant partition between Thaumarchaeota populations and warrants further analysis.
**4.4. Influences on the GDGT-IPL distribution along the Scotia Sea Transect**
Samples from the Scotia Sea were taken along a transect spanning 54 °S – 64 °S (Figure 1a). The T-S plot
(Figure 2) shows the CTD profiles for stations taken along the transect between 64 °S and 58.6 °S (CTD 3-
21) located south of the SBACC and stations between 53 °S and 58.6 °S (CTD 1, 22-25) located in the
Antarctic zone between the SBACC and the PF (Figure 1a). Results show IPL-GDGTs to be absent from the
mixed layer samples (15-40 m). Samples were collected between March and April 2012 and, similar to the
Amundsen Sea, the absence of IPL-GDGTs at these stations could be due to photo-inhibition and
competition from bacteria and algae at the surface (Church et al., 2003; Mincer et al., 2007; Hu et al., 2011;



Merbt et al., 2012; Luo et al., 2014). IPL-GDGTs are present at the surface CTD 1, 21, 20, and 16. CTD 21
and 20 are dominated by MH, which implies relic and not active archaeal populations (Lengger et al., 2013,
2014). CTD 16 contains low relative abundances of HPH-GDGT-0 which could indicate some archaeal
activity. However, CTD 1 contains greater IPL diversity including DH and HPH head groups potentially
suggesting a recently active archaeal community at the surface. CTD 1 is located close to the Falkland
Islands in the Subantarctic Zone north of the SAF and is potentially subject to additional terrestrial inputs
and coastal dynamics.
IPL-GDGTs were also found to be present within the thermocline (60-110 m) and contain a high proportion
of MH head group IPLs, suggesting a high proportion of relic IPL-GDGTs in the Scotia Sea that could relate
to variability in seasonality of archaeal populations. Further to this, DH-cren was found to be absent from the
thermocline, with HPH-cren intermittently present. This is consistent with our results from the Amundsen
Sea where DH-cren was mostly absent from the 120-240m depth intervals but present in the CDW depth
intervals (i.e. below 400m), while HPH-cren was present at both the thermocline and CDW depths.
As noted above, the Scotia Sea samples were collected along clear temperature (-1.6 to +7.3 °C), salinity
(33.6 -34.3 PSU), oxygen (218.3-332.7 ml L$^{-1}$), and fluorescence (0.03-1.1 ml m$^{-3}$) gradients associated with
ocean fronts, which are known to impact bacterioplankton population diversity (Wilkins et al., 2013; Baltar
et al., 2016; Raes et al., 2018). Figure 5 shows that higher latitude samples with cooler ocean temperatures
cluster positively on RDA axis 1 and have higher relative abundances of HPH-GDGT-0 and HPH-cren
(samples 3, 5, 7, 10, 13, 16, 18, 19), whilst samples from warmer ocean waters and lower latitudes cluster
negatively on RDA axis 1 and have higher relative abundances of MH and DH IPL-GDGTs (samples 20 –
24). This suggests that RDA 1 represents the transition across the SBACC. Temperature was found to be
statistically significant explanatory variable in the RDA which is consistent with previous research that has
identified clear links between core GDGT relative abundances and environmental variables such as
temperature (Schouten et al., 2007; Kim et al., 2008, 2010). Specifically, we observe a shift in the GDGT
head group between the warmer and cooler waters of the ACC fronts. Temperature, along with other
physicochemical properties (e.g. nutrient and oxygen concentrations) vary across the ACC (e.g. Rubin, 2003;
Freeman et al., 2019). These shifts in physicochemical properties across permanent oceanic boundaries
influence and control bacterial and archaeal species richness, creating ecological boundaries or niches (e.g.
Raes et al., 2018). Variability in IPL-GDGT headgroup composition observed across the Scotia Sea transect





could reflect the transition across an environmental niche (e.g. Elling et al., 2017; Bale et al., 2019). As this
study is limited by the number of chemical properties analysed, it would be speculative to infer the relative
importance of specific nutrient concentrations across the Scotia Sea transect.
Alternatively, the shift in IPL-GDGT head group could also be influenced by the presence of the Weddell
Gyre which is located south of 55-60 °S, and between 60 °W and 30 °E (Deacon et al., 1979). The Weddell
Gyre is a region of enhanced productivity, with austral summer chlorophyll A concentrations ranging from
1.5-10 µg/L (Bathmann et al., 1997; Cape et al. 2014) due to high concentrations of nutrients upwelled and
circulated through the gyre (Vernet et al., 2019 and references therein).
**5.  Conclusions**
A range of archaeal IPLs were detected in both the Amundsen Sea and the Scotia Sea. High relative
abundances of OH-GDGT core type were observed which could reflect the polar environmental setting of
these samples. Low cyclicity was detected for both the GDGT and OH-GDGT core type with acyclic OH-
GDGT-0 and GDGT-0, -1, -2, and cren reported.
In the Amundsen Sea a high relative abundance of IPL-GDGTs throughout the water column is indicative of
Thaumarchaeota activity both within the thermocline and CDW.  Indeed, the Thaumarchaeotal populations
within CDW could make a significant GDGT contribution to the sedimentary record which could have
implications for GDGT based temperature reconstructions.
IPL-GDGT relative abundance along the Scotia Sea transect shows a distinct pattern across the
oceanographic front transition. Samples south of the SBACC and from cooler ocean waters having higher
relative abundances of HPH-GDGT-0 and HPH-cren compared with samples north of the SBACC, and from
warmer ocean waters having higher relative abundances of MH and DH IPL-GDGTs. Indeed RDA reveals
that temperature is a significant explanatory variable, however, productivity and nutrient availability may
also play a role in IPL-GDGT distributions. Additionally, this shift in IPL-GDGT distributions could
represent a shift in the dominant archaeal IPL synthesisers and/or a physiological survival strategy.

Data availability
CTD data from JR257/JR272A are available from the British Oceanographic Data Centre at
https://www.bodc.ac.uk/data/documents/cruise/11431/.



Author contributions
CSJ, ELM, CDH, EM, JAS designed the experiments. CSJ, NJB, ECH, JM undertook the laboratory
preparation and analysis. PV, CA, TB, VP generated the oceanographic data. CSJ and AS undertook
statistical analysis. CSJ, ELM, NJB, ECH, SS, JAS wrote the manuscript with contributions from all authors.
Competing interests
The authors declare that they have no conflicting interests.
Acknowledgments
This project was funded through a UK Natural Environment Research Council (NERC) Standard Grant,
awarded to JS, ELM, CDH, and Kate Hendry (NE/M013081/1), a British Antarctic Survey Collaborative
Gearing Scheme award (ELM), a Helmholtz Research Grant (VH-NG-1101; JM), and the Durham
University Department of Geography Research Development Fund (CSJ). N.B. is funded by the European
Research Council (ERC) under the European Union's Horizon 2020 research and innovation program (grant
agreement no.694569). Collection of CTD casts on the A23 transect was supported by NERC National
Capability funding to BAS.  We thank M.D. West, A.J. Hayton, and D. Dorhout for technical support. We
are grateful to the captains, crews, support staff and scientists participating in cruises JR257, JR272 and
PS104, and acknowledge funding for cruise PS104 by AWI, MARUM, BAS and NERC UK-IODP.

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



Table 1: Scotia Sea SPM samples studied and their physical properties including sample depth (m) and
sample layer where "M" denotes mixed layer and "T" denotes thermocline layer (figures to 2 decimal places
(d.p.).

| Latitude (°N) | Longitude (°E) | Station | Sample Depth (m) | Layer | Temperature (°C) | Salinity (PSU) | Fluorescence (ml/m$^3$) |
|---|---|---|---|---|---|---|---|
| -53.013 | -58.04 | CTD 1 | 15 | M | 7.31 | 33.99 | 0.41 |
| -53.013 | -58.04 | CTD 1 | 100 | T | 6.12 | 34.03 | 0.13 |
| -53.586 | -42.835 | CTD 23 | 20 | M | 4.07 | 33.72 | 0.32 |
| -53.586 | -42.835 | CTD 23 | 100 | T | 2.23 | 33.81 | 0.08 |
| -52.88 | -41.787 | CTD 24 | 15 | M | 3.55 | 33.72 | 1.09 |
| -52.88 | -41.787 | CTD 24 | 80 | T | 1.67 | 33.88 | 0.09 |
| -53.743 | -38.155 | CTD 25 | 10 | M | 3.17 | 33.62 | 0.66 |
| -53.743 | -38.155 | CTD 25 | 80 | T | 1.95 | 33.91 | 0.05 |
| -57.119 | -31.815 | CTD 22 | 30 | M | 1.34 | 33.82 | 0.24 |
| -56.167 | -34.816 | CTD 22 | 110 | T | 0.84 | 34.12 | 0.09 |
| -57.459 | -31.327 | CTD 21 | 30 | M | 1.48 | 33.85 | 0.27 |
| -57.459 | -31.327 | CTD 21 | 110 | T | 1.34 | 34.3 | 0.03 |
| -57.803 | -30.83 | CTD 20 | 30 | M | 1.60 | 33.92 | 0.28 |
| -57.803 | -30.83 | CTD 20 | 110 | T | 1.01 | 34.15 | 0.06 |
| -58.213 | -30.822 | CTD 19 | 20 | M | 1.29 | 33.9 | 0.27 |
| -58.213 | -30.822 | CTD 19 | 80 | T | 1.16 | 34.19 | 0.09 |
| -58.624 | -30.821 | CTD 18 | 20 | M | 0.65 | 33.69 | 0.17 |
| -58.624 | -30.821 | CTD 18 | 90 | T | -0.83 | 33.99 | 0.17 |
| -59.436 | -30.861 | CTD 16 | 20 | M | -0.64 | 33.67 | 0.17 |
| -59.436 | -30.861 | CTD 16 | 70 | T | -1.32 | 34.12 | 0.08 |
| -60.319 | -30.961 | CTD 13 | 30 | M | -0.89 | 33.74 | 0.11 |
| -60.319 | -30.961 | CTD 13 | 65 | T | -1.16 | 34.01 | 0.11 |
| -61.171 | -31.045 | CTD 10 | 30 | M | -1.08 | 33.82 | 0.15 |
| -61.171 | -31.045 | CTD 10 | 80 | T | -1.08 | 34.23 | 0.11 |
| -62.084 | -31.174 | CTD 7 | 40 | M | -1.11 | 33.87 | 0.4 |
| -62.084 | -31.174 | CTD 7 | 75 | T | -1.54 | 34.33 | 0.16 |
| -62.784 | -30.706 | CTD 5 | 20 | M | -1.13 | 33.87 | 0.28 |
| -62.784 | -30.706 | CTD 5 | 70 | T | -1.49 | 34.34 | 0.14 |
| -63.346 | -29.569 | CTD 3 | 20 | M | -1.18 | 33.8 | 0.22 |
| -63.346 | -29.569 | CTD 3 | 60 | T | -1.58 | 34.31 | 0.21 |






Table 2: Amundsen Sea SPM samples studied and their physical properties (figures to 2 d.p)

| Latitude (° N) | Longitude (°E) | Station | Sample Depth (m) | Temperature (°C) | Salinity (PSU) | Fluorescence (ml/m$^3$) |
|---|---|---|---|---|---|---|
| -74.958 | -101.829 | PS104/003-1 | 10 | -0.72 | 33.96 | 0.48 |
| -74.958 | -101.829 | PS104/003-1 | 120 | -1.19 | 34.13 | 0.01 |
| -74.958 | -101.829 | PS104/003-1 | 180 | -1.23 | 34.17 | 0.01 |
| -74.958 | -101.829 | PS104/003-1 | 998 | 1.01 | 34.67 | -0.02 |
| -74.866 | -100.76 | PS104/007-1 | 20 | -0.12 | 33.52 | 3.78 |
| -74.866 | -100.76 | PS104/007-1 | 120 | -0.91 | 34.08 | 0.01 |
| -74.866 | -100.76 | PS104/007-1 | 240 | -1.33 | 34.14 | -0.01 |
| -74.866 | -100.76 | PS104/007-1 | 685 | 0.87 | 34.63 | -0.02 |
| -74.359 | -101.747 | PS104/017-1 | 10 | -0.17 | 33.42 | 7.89 |
| -74.359 | -101.747 | PS104/017-1 | 150 | -1.61 | 34.16 | 0.01 |
| -74.359 | -101.747 | PS104/017-1 | 1375 | 1.06 | 34.71 | -0.02 |
| -72.768 | -107.093 | PS104/022-1 | 10 | -0.59 | 33.13 | 1.09 |
| -72.768 | -107.093 | PS104/022-1 | 30 | -0.47 | 33.27 | 1.71 |
| -72.768 | -107.093 | PS104/022-1 | 120 | -1.54 | 34.1 | 0.07 |
| -72.768 | -107.093 | PS104/022-1 | 697 | 0.98 | 34.71 | -0.02 |
| -73.297 | -112.328 | PS104/043-2 | 10 | -1.34 | 32.82 | 1.51 |
| -73.297 | -112.328 | PS104/043-2 | 120 | -1.62 | 34.18 | 0.01 |
| -73.297 | -112.328 | PS104/043-2 | 454 | 0.15 | 34.51 | -0.02 |






Table 3: Relative abundances (%) and heat map of IPLs identified in Scotia Sea. Relative abundances >30%
indicated in red, low relative abundances <10% indicated in green. nd = not detected.

| CTD | Depth (m) | GDGT-0 MH | GDGT-0 DH | GDGT-0 HPH | GDGT-1 DH | Crenarchaeol MH | Crenarchaeol DH | Crenarchaeol HPH | OH-GDGT-0 MH | OH-GDGT-0 DH | OH-GDGT-0 HPH | diOH-GDGT-0 MH |
|---|---|---|---|---|---|---|---|---|---|---|---|---|
| 1 | 15 | 6.8 | nd | 49.6 | nd | 3.4 | nd | 18.6 | nd | 21.6 | nd | nd |
| 1 | 100 | 4.6 | nd | 54.9 | nd | 3.3 | nd | 5.6 | 2.6 | 28.2 | nd | 0.8 |
| 23 | 20 | nd | nd | nd | nd | nd | nd | nd | nd | nd | nd | nd |
| 23 | 100 | 31.0 | nd | nd | nd | 16.8 | nd | nd | 19.6 | 17.7 | nd | 14.9 |
| 24 | 15 | nd | nd | nd | nd | nd | nd | nd | nd | nd | nd | nd |
| 24 | 80 | 36.2 | nd | 1.6 | nd | 23.3 | nd | nd | 16.5 | 15.7 | nd | 6.7 |
| 25 | 10 | nd | nd | nd | nd | nd | nd | nd | nd | nd | nd | nd |
| 25 | 80 | 10.1 | 1.0 | 35.3 | nd | 6.1 | nd | 13.4 | 8.7 | 14.8 | 1.8 | 8.8 |
| 22 | 30 | nd | nd | nd | nd | nd | nd | nd | nd | nd | nd | nd |
| 22 | 110 | 13.5 | nd | 8.8 | nd | 11.9 | nd | nd | 21.7 | 23.7 | nd | 20.4 |
| 21 | 30 | 52.6 | nd | nd | nd | nd | nd | nd | 47.4 | nd | nd | nd |
| 21 | 110 | 9.3 | 4.0 | 10.2 | 3.5 | 4.5 | nd | nd | 11.8 | 35.3 | nd | 21.4 |
| 20 | 30 | 53.0 | nd | nd | nd | 24.5 | nd | nd | 22.5 | nd | nd | nd |
| 20 | 110 | 9.0 | nd | 31.8 | nd | 6.0 | nd | nd | 12.4 | 28.2 | nd | 12.6 |
| 19 | 20 | nd | nd | nd | nd | nd | nd | nd | nd | nd | nd | nd |
| 19 | 80 | 3.1 | nd | 55.7 | nd | 2.6 | nd | 4.8 | 6.4 | 19.2 | nd | 8.2 |
| 18 | 20 | nd | nd | nd | nd | nd | nd | nd | nd | nd | nd | nd |
| 18 | 90 | 4.2 | nd | 57.8 | nd | 1.9 | nd | 13.4 | 4.7 | 9.2 | 2.6 | 6.2 |
| 16 | 20 | nd | nd | 100.0 | nd | nd | nd | nd | nd | nd | nd | nd |
| 16 | 70 | 7.8 | nd | 45.9 | nd | 3.2 | nd | nd | 20.6 | 8.9 | nd | 13.6 |
| 13 | 30 | nd | nd | nd | nd | nd | nd | nd | nd | nd | nd | nd |
| 13 | 65 | 15.3 | nd | 54.2 | nd | 4.1 | nd | 11.1 | 10.5 | nd | nd | 4.8 |
| 10 | 30 | nd | nd | nd | nd | nd | nd | nd | nd | nd | nd | nd |
| 10 | 80 | 4.2 | nd | 82.6 | nd | 0.5 | nd | nd | 7.0 | nd | nd | 5.7 |
| 7 | 40 | nd | nd | nd | nd | nd | nd | nd | nd | nd | nd | nd |
| 7 | 75 | 7.2 | nd | 47.7 | nd | 2.5 | nd | nd | 29.8 | nd | nd | 12.7 |
| 5 | 20 | nd | nd | nd | nd | nd | nd | nd | nd | nd | nd | nd |
| 5 | 70 | 0.7 | nd | 71.1 | nd | 0.4 | nd | 16.3 | 2.3 | 4.8 | 2.5 | 1.9 |
| 3 | 20 | nd | nd | nd | nd | nd | nd | nd | nd | nd | nd | nd |
| 3 | 60 | 45.2 | nd | 22.7 | nd | 6.9 | nd | nd | 25.2 | nd | nd | nd |




Table 4: Relative abundances (%) and heat map of IPLs identified in Amundsen Sea. Relative abundances
>30% indicated in red, low relative abundances <10% indicated in green. nd = not detected.

| Station | Depth (cm) | GDGT-0 MH | GDGT-0 DH | GDGT-0 HPH | GDGT-1 DH | GDGT-2 DH | Crenarchaeol MH | Crenarchaeol DH | Crenarchaeol HPH | OH-GDGT-0 MH | OH-GDGT-0 DH | OH-GDGT-0 HPH | diOH-GDGT-0 MH |
|---|---|---|---|---|---|---|---|---|---|---|---|---|---|
| PS104/003-1 | 10 | 1.2 | nd | 81.8 | nd | nd | 0.2 | nd | 11.1 | 0.4 | 5.1 | nd | 0.2 |
| PS104/003-1 | 120 | 0.6 | 2.2 | 56.2 | 1.5 | nd | 0.3 | 0.1 | 11.7 | 4.9 | 16.5 | 0.5 | 5.5 |
| PS104/003-1 | 180 | 1.4 | nd | 18.0 | nd | nd | 0.7 | nd | nd | 24.1 | 25.7 | nd | 30.1 |
| PS104/003-1 | 998 | 3.4 | 11.3 | 28.1 | 14.7 | 8.2 | 1.7 | 3.0 | 4.3 | 5.2 | 18.8 | nd | 1.3 |
| PS104/007-1 | 20 | 89.1 | nd | nd | nd | nd | 10.9 | nd | nd | nd | nd | nd | nd |
| PS104/007-1 | 120 | 1.4 | 4.6 | 38.8 | 5.1 | 1.9 | 1.0 | 0.4 | 7.7 | 6.9 | 25.7 | nd | 6.5 |
| PS104/007-1 | 240 | 2.3 | 5.7 | 40.0 | 3.3 | nd | 1.3 | nd | 8.3 | 11.8 | 11.9 | nd | 15.4 |
| PS104/007-1 | 685 | 1.3 | 8.9 | 37.8 | 9.1 | 4.1 | 1.3 | 1.8 | 8.3 | 3.6 | 22.7 | nd | 1.1 |
| PS104/017-1 | 10 | nd | nd | nd | nd | nd | nd | nd | nd | nd | nd | nd | nd |
| PS104/017-1 | 150 | 1.7 | nd | 43.9 | nd | nd | 1.0 | nd | 6.8 | 14.1 | 13.0 | nd | 19.5 |
| PS104/017-1 | 1375 | 0.9 | 6.5 | 38.2 | 11.1 | 7.3 | 1.1 | 3.0 | 11.9 | 2.4 | 17.3 | nd | 0.3 |
| PS104/022-1 | 10 | nd | nd | nd | nd | nd | nd | nd | nd | nd | nd | nd | nd |
| PS104/022-1 | 30 | nd | nd | nd | nd | nd | nd | nd | nd | nd | nd | nd | nd |
| PS104/022-1 | 120 | 2.8 | nd | 51.6 | nd | nd | 1.7 | nd | 12.4 | 11.1 | 9.3 | 1.2 | 9.9 |
| PS104/022-1 | 697 | 4.3 | 6.0 | 31.5 | 11.2 | 5.3 | 2.0 | 2.3 | 5.6 | 5.5 | 25.0 | nd | 1.2 |
| PS104/043-2 | 10 | nd | nd | nd | nd | nd | nd | nd | nd | nd | nd | nd | nd |
| PS104/043-2 | 120 | 1.6 | nd | 38.3 | nd | nd | 0.5 | nd | 11.5 | 4.6 | 37.9 | 0.9 | 4.7 |
| PS104/043-2 | 454 | 0.7 | 0.2 | 72.3 | nd | nd | 0.2 | nd | 13.2 | 1.7 | 8.6 | 0.7 | 2.4 |






Table 5: Ratios of monohexose/dihexose (MH/DH), monohexose/hexosephosphohexose (MH/HPH), GDGT-
0/crenarchaeol (GDGT-0/Cren), and Ring Index in the Scotia Sea.

| Sea | Station | Depth (cm) | MH/DH | MH/HPH | GDGT-0/Cren | Ring Index |
|---|---|---|---|---|---|---|
| Scotia | CTD 1 | 15 | 0.5 | 0.1 | 2.6 | 0.9 |
| Scotia | CTD 1 | 100 | 0.4 | 0.2 | 6.7 | 0.4 |
| Scotia | CTD 23 | 20 | | | | |
| Scotia | CTD 23 | 100 | 4.7 | | 1.8 | 0.7 |
| Scotia | CTD 24 | 15 | | | | |
| Scotia | CTD 24 | 80 | 5.3 | 52.4 | 1.6 | 0.9 |
| Scotia | CTD 25 | 10 | | | | |
| Scotia | CTD 25 | 80 | 2.1 | 0.7 | 2.4 | 0.8 |
| Scotia | CTD 22 | 30 | | | | |
| Scotia | CTD 22 | 110 | 2.8 | 7.7 | 1.9 | 0.5 |
| Scotia | CTD 21 | 30 | | | | |
| Scotia | CTD 21 | 110 | 1.1 | 4.6 | 5.3 | 0.2 |
| Scotia | CTD 20 | 30 | | | 2.2 | 1.0 |
| Scotia | CTD 20 | 110 | 1.4 | 1.3 | 6.8 | 0.2 |
| Scotia | CTD 19 | 20 | | | | |
| Scotia | CTD 19 | 80 | 1.1 | 0.3 | 8.0 | 0.3 |
| Scotia | CTD 18 | 20 | | | | |
| Scotia | CTD 18 | 90 | 1.8 | 0.2 | 4.1 | 0.6 |
| Scotia | CTD 16 | 20 | | | | |
| Scotia | CTD 16 | 70 | 5.1 | 1.0 | 16.8 | 1.0 |
| Scotia | CTD 13 | 30 | | | | |
| Scotia | CTD 13 | 65 | | 0.5 | 4.6 | 0.6 |
| Scotia | CTD 10 | 30 | | | | |
| Scotia | CTD 10 | 80 | | 0.2 | 177.6 | 0.02 |
| Scotia | CTD 7 | 40 | | | | |
| Scotia | CTD 7 | 75 | | 1.1 | 21.7 | 0.1 |
| Scotia | CTD 5 | 20 | | | | |
| Scotia | CTD 5 | 70 | 1.1 | 0.1 | 4.3 | 0.7 |
| Scotia | CTD 3 | 20 | | | | |
| Scotia | CTD 3 | 60 | | 3.4 | 9.9 | 0.3 |






Table 6: Ratios of monohexose/dihexose (MH/DH), monohexose/hexosephosphohexose (MH/HPH),
GDGT-0/crenarchaeol (GDGT-0/Cren), and Ring Index in the Amundsen Sea.

| Sea | Station | Depth (cm) | MH/DH | MH/HPH | GDGT-0/Cren | Ring Index |
|------|---------|-----------|-------|--------|-------------|------------|
| Amundsen | PS104/003-1 | 10 | 0.4 | 0.0 | 7.3 | 0.5 |
| Amundsen | PS104/003-1 | 120 | 0.6 | 0.2 | 4.8 | 0.5 |
| Amundsen | PS104/003-1 | 180 | 2.2 | 3.1 | 27.0 | 0.03 |
| Amundsen | PS104/003-1 | 998 | 0.2 | 0.4 | 4.8 | 0.7 |
| Amundsen | PS104/007-1 | 20 | | | 8.2 | 0.4 |
| Amundsen | PS104/007-1 | 120 | 0.4 | 0.3 | 4.9 | 0.5 |
| Amundsen | PS104/007-1 | 240 | 1.5 | 0.6 | 5.0 | 0.4 |
| Amundsen | PS104/007-1 | 685 | 0.2 | 0.2 | 4.2 | 0.6 |
| Amundsen | PS104/017-1 | 10 | | | | |
| Amundsen | PS104/017-1 | 150 | 2.8 | 0.7 | 5.8 | 0.3 |
| Amundsen | PS104/017-1 | 1375 | 0.1 | 0.1 | 2.8 | 0.9 |
| Amundsen | PS104/022-1 | 10 | | | | |
| Amundsen | PS104/022-1 | 30 | | | | |
| Amundsen | PS104/022-1 | 120 | 2.8 | 0.4 | 3.8 | 0.6 |
| Amundsen | PS104/022-1 | 697 | 0.3 | 0.4 | 4.2 | 0.6 |
| Amundsen | PS104/043-2 | 10 | | | | |
| Amundsen | PS104/043-2 | 120 | 0.3 | 0.2 | 3.3 | 0.5 |
| Amundsen | PS104/043-2 | 454 | 0.6 | 0.1 | 5.4 | 0.5 |



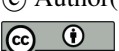

Figure 1. Map showing studied CTD sampling stations (red dots) in the Scotia sea (A) and Amundsen sea
(B). The main oceanic fronts are also shown in panel A; subantarctic (SAF), polar (PF), southern ACC
(SACCF) and the southern boundary of the ACC (SB) (Sokolov and Rintoul, 2009). The names of the ice
shelves are shown in panel B.

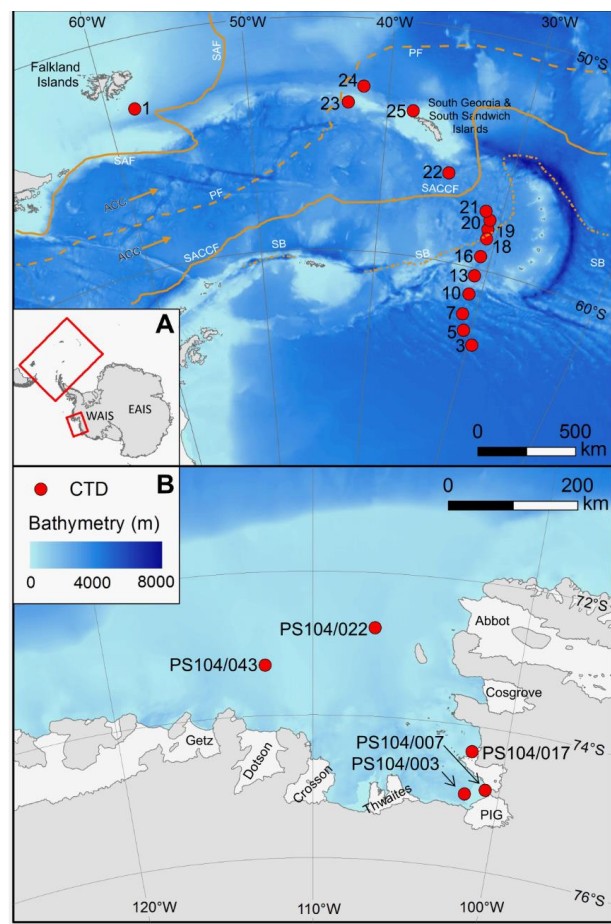




Figure 2. The Temperature Salinity profiles (T-S plot) for the Amundsen Sea (A) showing Antarctic Surface
Water (AASW) and Circumpolar Deep Water (CDW), and Scotia Sea (B). Black dots indicate the CTD data
and open triangles indicate the seawater sampling depths.

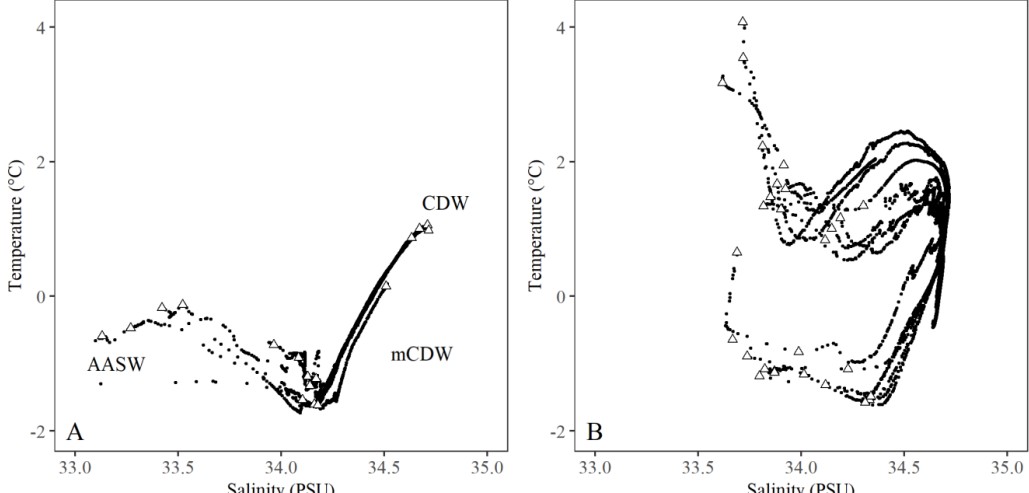




Figure 3. Total IPL-GDGT concentration (Log$_{10}$, units/L) (A) and relative abundance (%) of IPL head
groups, monohexose (MH, black), dihexose (DH, white), hexose-phosphohexose (HPH, grey) (B) in Scotia
Sea thermocline samples (mixed layer samples excluded from plots).

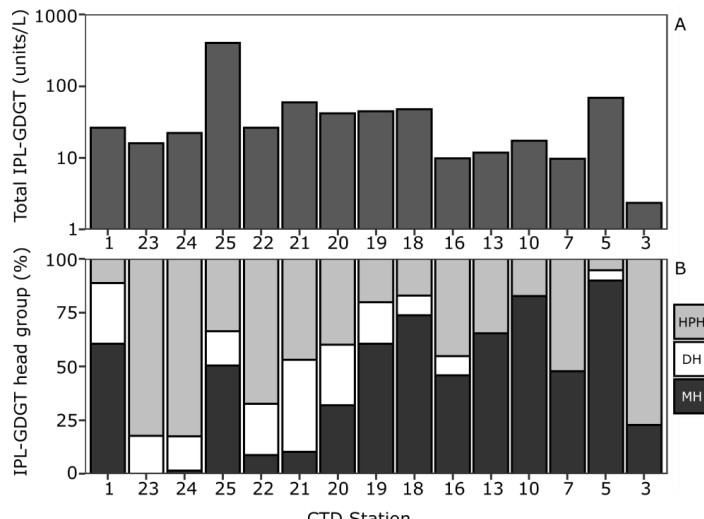






Figure 4. Relative abundance (%) of intact GDGTs at approximate sample depths in the Amundsen Sea. Bars reflect intact-GDGT head group with black representing MH head groups, white representing DH, and grey representing HPH. Contour lines show approximate ocean temperature ranges using CTD data taken at each sample station with Ocean Data View DIVA gridding.

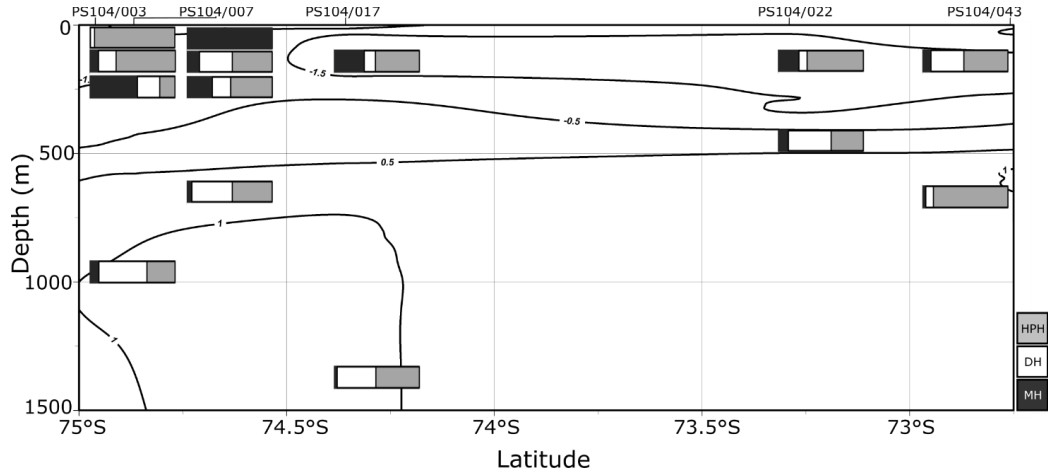



Figure 5. Redundancy analysis triplot for Scotia Sea sample set showing samples with depths, biomarker
response variables (grey lines), and explanatory variables (black with dashed lines indicating statistical
significance).

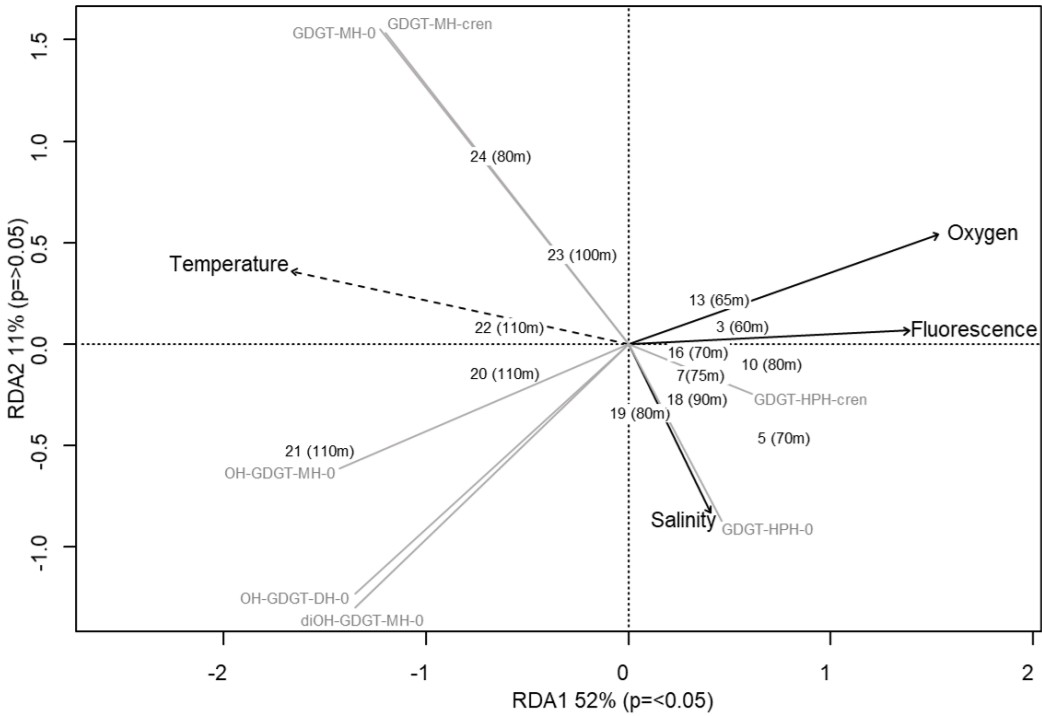




Supplement A. Absolute masses of IPLS detected in this study including for GDGTs, OH-GDGTs, and
diOH-GDGTs with either MH, DH, or HPH head groups, and for each adduct (H+, NH4+, and Na+).
Supplement B: S1. CTD matrix showing temperature (°C), salinity (PSU), Fluorescence (mg/m$^3$), Oxygen
(ml/L) for CTD stations PS104/003 (A), PS104/007 (B), PS104/017 (C), PS104/022 (D), PS104/043 (E),
with seawater sample depths indicated by a triangle.
S2. Intact GDGT structures showing GDGT cores where, GDGT: R & R' = H; OH-GDGT: R=OH, R'=H;
diOH-GDGT: R & R' = OH. Monohexose (MH), dihexose (DH), and hexose-phosphohexose (HPH) polar
head groups structures shown.
Supplement C. Redundancy analysis output for Scotia Sea sample set including ANOVA.