# Peer review of "Archaeal Intact Polar Lipids in Polar Waters: A Comparison Between"

_Biogeosciences, 2020_

## Referee Comment (RC1) · Anonymous Referee #1 · 15 Dec 2020

In this study, Spencer-Jones et al. present the results of intact polar lipid-GDGTs in suspended particulate matters from the Amundsen Sea and the Scotia Sea. The topic is interesting and relevant to the field of Biogeosciences, which is important for understanding and predicting the changes of West Antarctic Ice Sheet. The authors, however, are suggested to make the significance of the work clearer, and the discussion and conclusion more focused and concise. Thus, this manuscript is not ready for publication in the current stage for several reasons:

1. It is not clear what the outcome of the comparison between the Amundsen and Scotia Seas are. The authors mainly present the results in the two seas, respectively, but

rarely make comparisons between the two. Moreover, the methods of lipid extraction are different for the samples of two seas (line 171-187). The authors are suggested to make a reasonable discussion on how different methods of lipid extraction may influence the results.

2. It is important to understand the main driver of Southern Ocean GDGT distributions. However, there is a lack of strong relative discussion. For example, it is unclear how circumpolar deep water affects IPL distributions in Amundsen Sea (line 398-469). An improvement in this aspect would be helpful.

3. The authors are suggested to specify the main results in the Abstract. How did IPL-GDGT signatures shift in the study areas? And how did the signatures correlate with physicochemical parameters?

4. The structure of some sections should be revised. Please move the section 1.2 to Methodology and remove the subtitle of 1.1. Also, the Result section should be condensed by, for example, removing redundant sentences, e.g., repeating results on water masses (line 236 and 241).

5. The conclusion is suggested to be re-written to sufficiently show the significance of this study.

Some detailed suggestions are given below:

Line 109 and line 233: A colon should be used instead of semicolon.

Line 110: should it be SFACC?

Line 162-166: it is possible to quantify the proportion of 0.2-0.7 $\mu$m microbes?

Line 222-223: please check the use of semicolon: —- are not available, therefore–.

Line 238: please check the unit of oxygen concentration.

Line 283: Why to present the results of MH/DH and MH/HPH ratios? It is not introduced

in the Introduction or discussed later.

L288-293 and line 308-319: Please clarify to which figure or table these results refer.

L293-295: This statement is inconsistent with the information in Fig. 3b, which shows that the relative abundance of the HPH head group decreased from CTD station no. 1 to no. 3.

Line 302: Are samples 1 and 25 equal to CTD 1 and 25?

Line 332-333: this information is already presented in the last paragraph.

Line 349-350: repeated information, please revise this sentence.

Line 411: – have shown an increase in group I —

Line 413-418: are there any data of bacterial or algal community, and environmental factors from the same cruise? These data of biotic and abiotic factors would strongly support the discussion here.

Line 429: please specify what use of this head group as a biomarker for the archaeal community.

Line 430-431: Is this conclusion made based on the results from previous studies stated on line 423-425? If yes, how to exclude the possibility of the MH head group synthesized recently by archaea?

Line 438-442: are these data of productivity, nutrient concentration and light intensity from the same cruise with the present study?

Line 447-448: it is not clear how figure 4 shows an increase in diversity downwards in IPL-GDGT. As I see it, the diversity is also high in the surface layer as shown by bars.

Line 455 and line 498: what does the word 'This' exactly mean? Please clarify.

Line 466-467: the difference between stations?

Line 468: It is not clear what the small contrast in HPH and DH-cren distribution is.

Line 471-474: These sentences should be moved to the Method or Result section.

Line 516: — was detected —

Fig. 1: This figure should be improved. For example, the color and size of the names of ocean fronts should be changed. Fig. 2: Consider to use different colors for different water masses and write the Amundsen Sea and the Scotia Sea in the panel. Fig. 3: It is not clear why GDGT concentration is presented as units/L.

---

## Referee Comment (RC2) · Anonymous Referee #2 · 23 Dec 2020

This study attempts to assess the distribution of core GDGTs and intact polar lipid (IPL)-GDGTs in the waters of the Scotia Sea and Amundsen Sea in the Southern Ocean. GDGTs are membrane spanning isoprenoidal lipids that make up a significant portion of the membrane bilayer of a variety of Archaea. Modifications to these core membrane lipids, including the addition of 1-8 cyclopentane rings, 1 cyclohexane ring, or hydroxylations, or changes to the polar head groups, are thought to represent physiological responses to environmental factors, such as changes in temperature, pH, or redox. Because of this physiological connection between GDGT modifications and environmental conditions, an because GDGTs can be well preserved in ancient sediments, these molecules have been employed as paleotemperature proxies. In addition,

some of these modified GDGTs have also been proposed to be restricted to certain archaeal groups and, thus, have been utilized as diagnostic markers for specific archaeal group in a environmental settings. However, the utility of GDGTs and IPL-GDTGs as proxies and/or diagnostic markers requires an understanding of a variety of factors – confirming the correlation between environmental factors and the specific modifications made on the GDGT structures, determining the distribution of various GDGT structures in different ecosystems, and assessing the occurrence of specific GDGT structures in different cultured archaeal groups. In this study, the authors investigate the occurrence of the core GDGT structures, which are most relevant for paleotemperature proxies, as well as the occurrence of the GDGT structures with various polar headgroups in the Scotia Sea and Amundsen Sea.

The analyses performed in this study are well done and provide an interesting picture of the distribution of GDGTs in the Southern Ocean. In particular, they show limited cyclization of GDGTS in their samples with the majority of core GDGTs having zero rings. In addition, they see a significant amount of hydroxylated GDGTs which have been proposed to function in helping maintain membrane fluidity at low temperatures. The authors infer that both observations may reflect the cold environment of the Southern Ocean which their specific sites can range from -1 to 8 degrees Celsius. The occurrence of IPLs is a little more difficult to parse. Although I agree that IPLs can represent the occurrence of living archaea in the water column, I am not convinced that the intact IPLs are useful as diagnostic markers specifically for the Thaumarchaeota as I believe other archaea are known to produce head groups with 1 or 2 hexose groups. Nonetheless, the authors are able to demonstrate some interesting IPL-GDGT patterns that may reflect temporal changes in archaeal communities. For example, in the surface samples form the Amundsen Sea (collected within the euphotic zone), there was an absence of IPL-GDGTs. Previous studies have shown the absence of archaea in the surface waters of the Southern Ocean (and large abundance of bacteria) and this lack of IPL-GDGTs corresponded well with that larger seasonal variation in archaeal populations. Overall, this study is well-designed and well-written and contributes some

significant knowledge into the environmental distribution of both core GDGTs and IPL-GDGTs.

---

## Author Comment (AC1) · 28 Jan 2021

On behalf of the co-authors, I thank this anonymous referee for their helpful comments and the time they took to improve the paper. We reply to the reviewer's comments in bold below, where line numbers refer to those in our original submission.

*In this study, Spencer-Jones et al. present the results of intact polar lipid-GDGTs in suspended particulate matters from the Amundsen Sea and the Scotia Sea. The topic is interesting and relevant to the field of Biogeosciences, which is important for understanding and predicting the changes of West Antarctic Ice Sheet. The authors, however, are suggested to make the significance of the work clearer, and the discussion and conclusion more focused and concise. Thus, this manuscript is not ready for publication in the current stage for several reasons:*

*1. It is not clear what the outcome of the comparison between the Amundsen and Scotia Seas are. The authors mainly present the results in the two seas, respectively, but rarely make comparisons between the two.*

**In this study the two sites are used together to discuss the occurrence of Hydroxylated GDGTs and the diversity in cyclopentane rings in polar waters. This discussion happens from lines 322 – 347 (section 4.1). Our results of high relative abundance of hydroxylated GDGTs and low abundance of cyclopentane-containing GDGTs within both the Amundsen Sea and Scotia Sea suggests that these GDGTs distributions may be common throughout the Southern Ocean (Sections 4.1 and 4.2, where the data are discussed together). Furthermore, these observations from the Amundsen and Scotia Seas are notably different from other published IPL-GDGT data from non-polar settings (e.g. Besseling et al., 2019). We then discuss the features of the individual data sets (e.g. the influence of water masses on IPL-GDGT distributions; sections 4.3 and 4.4).**

*1 continued Moreover, the methods of lipid extraction are different for the samples of two seas (line 171-187). The authors are suggested to make a reasonable discussion on how different methods of lipid extraction may influence the results.*

**We comment that the extraction methods are different between the Scotia Sea and the Amundsen Sea in lines 184-187. The method used for the extraction of the Amundsen Sea samples is not the Bligh Dyer protocol most commonly used for IPL extraction. Extraction technique has not been found to significantly affect c-GDGT recovery (Schouten et al., 2013; Weber et al., 2017) but has been found to have a greater influence on IPL-GDGT recovery due to differences in polar moieties (Weber et al., 2017). Previous analysis of SPM from Lake Lugano determined the impact of different extraction protocols (Bligh and Dyer and Ultrasonic extraction) on IPL-GDGT recovery and quantification (Weber et al., 2017). Weber et al. (2017) found the extraction procedure impacts the absolute quantification of GDGTs along with the recovery of cren' (under-quantified) and GDGT-3 (over-quantified). Sample purification using silica gel column chromatography has also been found to have an impact on IPL-GDGT recovery (Pitcher et al., 2009; Lengger et al., 2012) with highest IPL-GDGT recoveries achieved with methanol as the final eluent (Pitcher et al., 2009). Hexose-Phosphohexose-GDGTs (HPH-GDGTs) can be significantly impacted by silica gel chromatography with peak areas of HPH-GDGTs up to 80% smaller compared with untreated IPL-GDGT extracts (Lengger et al., 2012).**

**In our study we observe lower concentrations of total IPL-GDGTs in the Amundsen Sea compared to the Scotia Sea (Figure 1A and 1B), this is consistent with previous studies analysing extraction and purification methodologies (e.g. Lengger et al., 2012; Weber et al., 2017). There may also be biases in the recovery of HPH-GDGTs compared with dihexose- (DH) and monohexose-GDGTs (MH). Figure 1C shows the ratio of MH, DH, and HPH head groups. The ratio of MH/DH in the Amundsen and Scotia seas is similar, and this would be consistent with previous research suggesting MH and DH head groups are less biased by chromatographic purification**

methodology (Lennger et al., 2012). Previous research has identified HPH-GDGTs to be more impacted by work up techniques. However, HPH-GDGTs were identified in the Amundsen sea as a significant component of the IPL-GDGT suite (e.g. Figure 4 of original submitted manuscript). This suggests that re-analysis of SPM from the Amundsen sea would yield greater concentrations of IPL-GDGTs and potentially even higher relative abundances of HPH-GDGT relative to MH and DH.

[Figure]

Figure 1. A: Boxplot showing semi-quantitative concentrations of total IPL-GDGTs (units/L) in the Amundsen Sea and Scotia Sea (high outlier of 200 units/L not shown). B: Boxplots showing semi-quantitative concentrations (units/L) of IPL-GDGT headgroups, monohexose (MH-GDGTs), dihexose (DH-GDGT), and hexose-phosphohexose (HPH-GDGT) in the Amundsen (white) and Scotia (grey) sea. C: Boxplot showing the ratio between IPL-GDGT headgroups including, MH:DH, DH:HPH and MH:HPH in the Amundsen (white) and Scotia (grey) seas. Samples where IPL-GDGTs were below detection limit of instrument are excluded from analysis. Black circles indicate outliers, white triangles indicate mean values, black line indicates median values.

We acknowledge that there may be some differences in IPL-GDGT recovery between the Amundsen and Scotia sea samples due to differences in extraction and work-up technique. However, we propose that comparison can still be made between the two seas as we do not report absolute quantities of IPL-GDGTs as the methods are semi-quantitative, we do not report the occurrence of cren', and GDGT-3 was below the detection limit of the instrument. The exceptionally high relative abundance of HPH-GDGTs in the Amundsen Sea suggests that our study does characterise the IPL-GDGT suite. Furthermore, our observations of active IPL-GDGT synthesis throughout the water column in the Amundsen sea are still relevant despite these biases in sample work-up techniques.
We propose adding some extra comments to this in the methodology section in a revised manuscript.

*2. It is important to understand the main driver of Southern Ocean GDGT distributions. However, there is a lack of strong relative discussion. For example, it is unclear how circumpolar deep water affects IPL distributions in Amundsen Sea (line 398-469). An improvement in this aspect would be helpful.*
We acknowledge that some of the significance of this study is not discussed, therefore, we will restructure the discussion, separating out the surface trends from the CDW trends. Furthermore, we will expand the discussion of the CDW trends to take into account the reviewers concern.
We propose adding the following text to the discussion of CDW in section 4.3.
"Thaumarchaeota have been observed to live throughout the water column, however, the distribution of Thaumarchaeota is not homogenous. Molecular microbiology has identified Thaumarchaeota to be virtually absent from Antarctic Summer Surface Waters (0-45m depth) and present in Winter Water (45-105m depth) and Circumpolar Deep Water (105-3500m depth) (e.g. Kalanetra et al., 2009). Indeed, temperature reconstructions based on GDGTs are suggested to represent more specifically 45-200m range (Kim et al., 2012), acknowledging the absence of Thaumarchaeota from the surface waters during the summer months in Antarctica. The influence of CDW on reconstructed TEX86 paleo temperatures has been hypothesised at Adélie Land (East Antarctica) with Kim et al., 2012 suggesting warmer reconstructed temperatures likely

due to the upwelling of CDW onto the piston core site. In our study we specifically observe intact polar lipid GDGTs of recently living archaea in the circumpolar deep water (over 500 m water depth). Furthermore, we observe a shift in head group composition at CDW depths in the Amundsen sea representing a shift in the IPL-GDGT producing community. We hypothesise that the contribution of GDGTs synthesised at CDW depths where physical parameters (e.g. temperature) can be strikingly different to the 45-200m water depth may have a significant impact on reconstructed TEX86 temperatures, not only the Amundsen sea but potentially more broadly within the Southern Ocean."

*3. The authors are suggested to specify the main results in the Abstract. How did IPL-GDGT signatures shift in the study areas? And how did the signatures correlate with physicochemical parameters?*
**We will update the abstract to reflect more details of the results of the GDGT distributions including the high relative abundance of hydroxylated GDGTs, low cyclic diversity and the implications of active GDGT synthesis in the CDW.**

*4. The structure of some sections should be revised. Please move the section 1.2 to Methodology and remove the subtitle of 1.1. Also, the Result section should be condensed by, for example, removing redundant sentences, e.g., repeating results on water masses (line 236 and 241).*
**We will edit the results section according to the comments made by Reviewer 1, to improve readability of the manuscript**

*5. The conclusion is suggested to be re-written to sufficiently show the significance of this study.*
**We will edit the conclusions to highlight the main findings of our paper. For example, underlined sections indicate new sentences that we will add.**
**"A range of archaeal IPLs were detected in both the Amundsen Sea and the Scotia Sea. High relative abundances of OH-GDGT core type were observed which could reflect the polar environmental setting of these samples. Low cyclicity was detected in both the Amundsen and Scotia Seas for both the GDGT and OH-GDGT core type with acyclic OH-GDGT-0 and GDGT-0, -1, -2, and cren reported. Low cyclicity of GDGTs may potentially be a more widespread feature of the Southern Ocean GDGT signature.**

**IPL-GDGT relative abundance along the Scotia Sea transect shows a distinct pattern across the oceanographic front transition. Samples south of the SBACC and from cooler ocean waters having higher relative abundances of HPH-GDGT-0 and HPH-cren compared with samples north of the SBACC, and from warmer ocean waters having higher relative abundances of MH and DH IPL-GDGTs. Indeed, RDA reveals that temperature is a significant explanatory variable, however, productivity and nutrient availability may also play a role in IPL-GDGT distributions. Additionally, this shift in IPL-GDGT distributions could represent a shift in the dominant archaeal IPL synthesisers and/or a physiological survival strategy."**

**In the Amundsen Sea IPL-GDGTs are detected throughout the water column. IPL-GDGTs of recently living archaea were specifically observed in the circumpolar deep water (over 500 m water depth) along with a shift in head group composition at CDW depths representing a change in the IPL-GDGT producing community. We hypothesise that the contribution of GDGTs synthesised at CDW depths where physical parameters, such as temperature, can be strikingly different to the upper water column (e.g. 0-200m water depth) may have a significant impact on**

**reconstructed TEX86L temperatures in not only the Amundsen sea but potentially more broadly within the Southern Ocean.**

*Some detailed suggestions are given below:*
*Line 109 and line 233: A colon should be used instead of semicolon.*
***This will be corrected***

*Line 110: should it be SFACC?*
**SAACF is a widely used abbreviation for the Southern Front of the ACC both within the Antarctic research field and within the Biogeosciences Journal.**

*Line 162-166: it is possible to quantify the proportion of 0.2-0.7 _m microbes?*
**Unfortunately it is not possible to quantify the number of microbes collected on the filters at the time of sampling.**

*Line 222-223: please check the use of semicolon: —- are not available, therefore–.*
**This will be corrected**

*Line 238: please check the unit of oxygen concentration.*
**We have corrected this typo to µmol/Kg.**

*Line 283: Why to present the results of MH/DH and MH/HPH ratios? It is not introduced in the Introduction or discussed later.*
**We present these data as ratios as an additional way to explore the data. As MH is predominantly a degradation product, we thought it could be interesting to see how the ratio of this headgroup changes with the more labile headgroups. However, as these do not form part of the main discussion in the paper we will remove reference to them to improve readability.**

*L288-293 and line 308-319: Please clarify to which figure or table these results refer.*
**Lines 288-293 reference table 3 and Fig. 3**
**Lines 308-319 reference Fig. 5 and Supplement C, however, we will edit supplement C to include the correlation coefficients.**
**We will edit the text to reflect this.**

*L293-295: This statement is inconsistent with the information in Fig. 3b, which shows that the relative abundance of the HPH head group decreased from CTD station no. 1 to no. 3.*
**The MH and HPH in the legend of Figure 3b had been mislabelled. We have now corrected the legend (see Figure 2 below) and therefore, the results are now consistent.**

[Figure]

**Figure 2.** Total IPL-GDGT concentration (Log$_{10}$, units/L) (A) and relative abundance (%) of IPL head groups, monohexose (MH, grey), dihexose (DH, white), hexose-phosphohexose (HPH, black) (B) in Scotia Sea thermocline samples (mixed layer samples excluded from plots). Revised figure 3.

*Line 302: Are samples 1 and 25 equal to CTD 1 and 25?*
**Yes: this will be clarified in text**

*Line 332-333: this information is already presented in the last paragraph.*
**This will be corrected**

*Line 349-350: repeated information, please revise this sentence.*
**This will be corrected**

*Line 411: – have shown an increase in group I ——*
**This will be corrected**

*Line 413-418: are there any data of bacterial or algal community, and environmental factors from the same cruise? These data of biotic and abiotic factors would strongly support the discussion here.*
**We agree that bacterial/algal community data would improve the discussion. However, this data was not collected (beyond the CTD measurements) for either the Scotia Sea or Amundsen Sea cruises, since these expeditions were focussed on palaeoceanography and ice sheet history, respectively.**

*Line 429: please specify what use of this head group as a biomarker for the archaeal community.*
**Our statement referred to DH-cren being a potential marker for the activity of the archaeal community (i.e. high DH-cren could mean an abundance of ammonia oxidising archaea) rather than as a biomarker *proxy* for an environmental variable. We will restructure some of the text to bring the content of lines 429 up to lines 458 and improve these sentences.**
**"HPH-cren, however, may not be the most suitable proxy for tracking the complete AOA population as the relative abundance of this IPL can vary significantly between phylogenetic subgroups (Elling et al., 2017). While DH head group is not as labile as HPH due to its glycosidic structure (Lengger et al., 2013), DH-GDGTs have been identified with consistent relative abundances across the Nitrosopumilales order**

**(Group 1.1a), suggesting DH-cren as an additional biomarker for AOA activity (Elling et al., 2017)."**

*Line 430-431: Is this conclusion made based on the results from previous studies stated on line 423-425? If yes, how to exclude the possibility of the MH head group synthesized recently by archaea?*
**The conclusion in lines 430-431 is based on the previous work by Lengger et al. (2013, 2014). At PS104/007 we only observe the monohexose (MH) head group. Previously analysed IPL-GDGT profiles of marine mesophiles (e.g. Elling et al., 2017; Sinninghe Damsté et al., 2012; Bale et al., 2019) report MH-GDGT co-occurring with HPH and DH (Elling et al., 2017; Sinninghe Damsté et al., 2012; Bale et al., 2019). Therefore, if the MH-GDGTs detected at PS104/007 were from a recently active archaeal population then we would expect to also observe other IPL-GDGTs (e.g. HPH-GDGTs). The results from the surface of PS104/007 are in contrast with PS104/003 and Scotia sea CTD 1 where MH co-occurs with DH and HPH, suggesting active archaeal populations in the surface water depths.**

*Line 438-442: are these data of productivity, nutrient concentration and light intensity from the same cruise with the present study?*
**No nutrient data is available for the cruise where our samples were collected. We will edit the text to clarify that these data referenced are not from the same cruise.**

*Line 447-448: it is not clear how figure 4 shows an increase in diversity downwards in IPL-GDGT. As I see it, the diversity is also high in the surface layer as shown by bars.*
**We acknowledge that figure does not demonstrate our point that IPL-GDGT diversity increases with depth as we only show the head group. This point is better demonstrated by Table 4 and therefore we will update the text to reflect this. We observe an increase in the number of different IPL-GDGTs detected with depth as show in Figure 3 below.**

[Figure]

**Figure 3. Number of different IPL-GDGT compounds detected at each station in the Amundsen Sea (PS104/003, PS104/007, PS104/017, PS104/022, PS104/043) at each sample depth from the sea surface (m).**

*Line 455 and line 498: what does the word 'This' exactly mean? Please clarify.*
**We will edit and clarify.**

*Line 466-467: the difference between stations?*
**This sentence will be edited and clarified to indicate that we are discussing differences in depth.**

*C3Line 468: It is not clear what the small contrast in HPH and DH-cren distribution is.*
**This sentence will be edited and clarified with 466-467.**

*Line 471-474: These sentences should be moved to the Method or Result section.*
**This will be corrected**

*Line 516: — was detected —*
**This will be corrected**

*Fig. 1: This figure should be improved. For example, the color and size of the names of ocean fronts should be changed. Fig. 2: Consider to use different colors for different water masses and write the Amundsen Sea and the Scotia Sea in the panel. Fig. 3: It is not clear why GDGT concentration is presented as units/L.*
**We will edit these figures following the reviewer's suggestions.**

**Figure 3: GDGT concentration is presented at units/L as the method is semiquantitative (as mentioned in lines 222-224), despite this concentration can be compared between samples but should not be considered absolute. Therefore, we omit the units to prevent misleading the reader. This approach has been used in other IPL-GDGT publications (e.g. Besseling et al., 2019).**

References Used in Discussion

Bale, N.J., Palatinszky, M., Rijpstra, W.I.C., Herbold, C.W., Wagner, M. & Sinninghe Damsté, J.S.: Membrane lipid composition of the moderately thermophilic ammonia-oxidizing archaeon "Candidatus Nitrosotenuis uzonensis" at different growth temperatures. 85, 1-17, http:// doi.org/10.1128/AEM.01332-19, 2019.

Besseling, M.A., Hopmans, E.C., Koenen, M., van der Meer, M.T.J., Vreugdenhil, S., Schouten, S., Sinninghe Damsté, J.S. & Villanueva, L.: Depth-related differences in archaeal populations impact the isoprenoid tetraether lipid composition of the Mediterranean Sea water column. Org. Geochem. 135, 16-31, http://doi.org/10.1016/j.orggeochem.2019.06.008. 2019.

Elling, F.J., Konneck, M., Nicol, G.W., Stieglmeier, M., Bayer, B., Spieck, E., de la Torre, J.R., Becker, K.W., Thomm, M., Prosser, J.I., Herndl, G.J., Schleper, C. & Hinrichs, K.U.: Chemotaxonomic characterisation of the thaumarchaeal lipidome. Environ. Microbiol. 19, 2681-2700, http://doi.org/10.1111/1462-2920.13759, 2017.

Kalanetra, K.M., Bano, H. & Hollibaugh, J.T.: Ammonia-oxidizing Archaea in the Arctic Ocean and Antarctic coastal waters. Environ. Microbiol. 11, 2434-2445, http://doi.org/10.1111/j.1462-2920.2009.01974.x, 2009.

Kim, J.H., Crosta, X., Willmott, V., Renssen, H., Bonnin, J., Helmke, P., Schouten, S. & Sinninghe Damsté.: Holocene subsurface temperature variability in the eastern Antarctic continental margin. Geophys. Res. Lett. 39, 1-6, http:// doi.org/10.1029/2012gl051157, 2012.

Lengger, S.K., Hopmans, E.C., Sinninhe Damsté, J.S. & Schouten, S.: Comparison of extraction and work up techniques for analysis of core and intact polar tetraether lipids from sedimentary environments. Org. Geochem. 47, 34-40, http://doi.org/10.1016/j.orggeochem.2012.02.009, 2012.

Lennger, S.K., Kraaij,M., Tjallingii, R., Baas, M., Stuut, J.B., Hopmans, E.C., Sinninghe Damsté, J.S. & Schouten, S.: Differential degradation of intact polar and core glycerol dialkyl glycerol tetraether lipids upon post-depositional oxidation. Org. Geochem. 65, 83-93, https://doi.org/10.1016/j.orggeochem.2013.10.004, 2013.

Lennger, S.K., Hopmans, E.C., Sinninghe Damsté, J.S. & Schouten, S.: Fossilization and degradation of archaeal intact polar tetraether lipids in deeply buried marine sediments (Peru Margin). Geobiology. 12, 212-220, http:// doi.org/10.1111/gbi.12081, 2014.

Schouten, S., Hopmans, E.C., Rosell-Melé, A., Pearson, A., Adam, P., Bauersachs, T., Bard, E., Bernasconi, S.M., Bianchi, T.S., Brocks, J.J., Carlson, L.T., Castañeda, I.S., Derenne, S., Selver, A.D., Dutta, K., Eglinton, T., Fosse, C., Galy, V., Grice, K., Hinrichs, K.U., Huang, Y., Huguet, A., Huguet, C., Hurley, S., Ingalls, A., Jia, G., Keely, B., Knappy, C., Kondo, M., Krishnan, S., Lincoln, S., Lipp, J., Mangelsdorf, K., Martínez-García, A., Ménot, G., Mets, A., Mollenhauer, G., Ohkouchi, N., Ossebaar, J., Pagani, M., Pancost, R.D., Pearson, E.J., Peterse, F., Reichart, G.J., Schaeffer, P., Schmitt, G., Schwark, L., Shah, S.R., Smith, R.W., Smittenberg, R.H., Summons, R.E., Takano, Y., Talbot, H.M., Taylor, K.W.R., Tarozo, R., Uchida, M., Van Dongen, B.E., Van Mooy, B.A.S., Wang, J., Warren, C., Weijers, J.W.H., Werne, J.P., Woltering, M., Xie, S., Yamamoto, M., Yang, H., Zhang, C.L., Zhang, Y., Zhao, M. & Sinninghe Damsté, J.S.: An interlaboratory study of TEX86 and BIT analysis of sediments, extracts, and standard mixtures, Geochem. Geophys. Geosyst., 14, 5263-5265, https://doi.org/10.1002/2013GC004904, 2013.

Sinninghe Damsté, J.S., Rijpstra, W.I.C., Hopmans, E.C., Jung, M.Y., Kim, J.G., Rhee, S.K., Stieglmeier, M. & Schleper, C.: Intact Polar and Core Glycerol Dibiphytanyl Glycerol Tetraether Lipids of Group I.1a and I.1b Thaumarchaeota in Soil. 78, 6866-6874, http://doi.org/10.1128/aem.01681-12, 2012.

Weber, Y., Sinninghe Damsté, J.S., Hopmans, E.C., Lehmann, M.F. & Niemann, H.: Incomplete recovery of intact polar glycerol dialkyl glycerol tetraethers from lacustrine suspended biomass. Limn. Oceanogr. Methods. 15, 782-793. http://doi.org/10.1002/lom3.10198, 2017.

---

## Author Response (AR2)

**BG-2020-333 *Archaeal Intact Polar Lipids in Polar Waters: A Comparison Between the Amundsen and Scotia Seas***

**Reply to Editor**

We thank the editor for their final comments on the manuscript. As suggested we have:

1. Changed the units of oxygen from µmol/kg to µmol kg$^{-1}$ throughout the manuscript and supplementary figure B-S1.
2. We have corrected the typo on Table 3.
3. We have re-formatted Tables 3 and 4 to an accessible colour format.

**BG-2020-333** *Archaeal Intact Polar Lipids in Polar Waters: A Comparison Between the Amundsen and Scotia Seas*

**Reply to Editor**

This document combines the published responses we made to our two reviewers with updated line numbers that reference the changes we have made to the manuscript (tracked changed version).

In addition to addressing the reviewer's comments related to restructuring the manuscript we have adjusted the figure and table numbering. These edits are all visible in the "tracked changes" document.

**BG-2020-333** *Archaeal Intact Polar Lipids in Polar Waters: A Comparison Between the Amundsen and Scotia Seas*

**Authors comments**

We thank both reviewers for their positive reviews and constructive comments. We have replied to each of their concerns in the accompanying replies.

Line numbers refer to the tracked changed manuscript. Figure and table numbers have been updated during editing and these are noted where applicable in the replies to the reviewers.

**BG-2020-333** *Archaeal Intact Polar Lipids in Polar Waters: A Comparison Between the Amundsen and Scotia Seas*

**Reply to Reviewer 1**

On behalf of the co-authors, I thank this anonymous referee for their helpful comments and the time they took to improve the paper. We reply to the reviewer's comments in bold below, where line numbers refer to those in our tracked changed submission.

*In this study, Spencer-Jones et al. present the results of intact polar lipid-GDGTs in suspended particulate matters from the Amundsen Sea and the Scotia Sea. The topic is interesting and relevant to the field of Biogeosciences, which is important for understanding and predicting the changes of West Antarctic Ice Sheet. The authors, however, are suggested to make the significance of the work clearer, and the discussion and conclusion more focused and concise. Thus, this manuscript is not ready for publication in the current stage for several reasons:*

*1. It is not clear what the outcome of the comparison between the Amundsen and Scotia Seas are. The authors mainly present the results in the two seas, respectively, but rarely make comparisons between the two.*
**In this study the two sites are used together to discuss the occurrence of Hydroxylated GDGTs and the diversity in cyclopentane rings in polar waters. This discussion happens from lines 393-419 (section 4.1). Our results of high relative abundance of hydroxylated GDGTs and low abundance of cyclopentane-containing GDGTs within both the Amundsen Sea and Scotia Sea suggests that these GDGTs distributions may be common throughout the Southern Ocean (Sections 4.1-4.3, where the data are discussed together). Furthermore, these observations from the Amundsen and Scotia Seas are notably different from other published IPL-GDGT data from non-polar settings (e.g. Besseling et al., 2019). We then discuss the features of the individual data sets (e.g. the influence of water masses on IPL-GDGT distributions; sections 4.4 and 4.5).**

*1 continued Moreover, the methods of lipid extraction are different for the samples of two seas (line 171-187). The authors are suggested to make a reasonable discussion on how different methods of lipid extraction may influence the results.*
**The method used for the extraction of the Amundsen Sea samples is not the Bligh Dyer protocol most commonly used for IPL extraction. Extraction technique has not been found to significantly affect c-GDGT recovery (Schouten et al., 2013; Weber et al., 2017) but has been found to have a greater influence on IPL-GDGT recovery due to differences in polar moieties (Weber et al., 2017). Previous analysis of SPM from Lake Lugano determined the impact of different extraction protocols (Bligh and Dyer and Ultrasonic extraction) on IPL-GDGT recovery and quantification (Weber et al., 2017). Weber et al. (2017) found the extraction procedure impacts the absolute quantification of GDGTs along with the recovery of cren' (under-quantified) and GDGT-3 (over-quantified). Sample purification using silica gel column chromatography has also been found to have an impact on IPL-GDGT recovery (Pitcher et al., 2009; Lengger et al., 2012) with highest IPL-GDGT recoveries achieved with methanol as the final eluent (Pitcher et al., 2009). Hexose-Phosphohexose-GDGTs (HPH-GDGTs) can be significantly impacted by silica gel chromatography with peak areas of HPH-GDGTs up to 80% smaller compared with untreated IPL-GDGT extracts (Lengger et al., 2012). In our study we observe lower concentrations of total IPL-GDGTs in the Amundsen Sea compared to the Scotia Sea (Figure 1A and 1B), this is consistent with previous studies analysing extraction and purification methodologies (e.g. Lengger et al., 2012; Weber et al., 2017). There may also be biases in the recovery of HPH-GDGTs compared with dihexose- (DH) and monohexose-GDGTs (MH). Figure 1C shows the**

ratio of MH, DH, and HPH head groups. The ratio of MH/DH in the Amundsen and Scotia seas is similar, and this would be consistent with previous research suggesting MH and DH head groups are less biased by chromatographic purification methodology (Lennger et al., 2012). Previous research has identified HPH-GDGTs to be more impacted by work up techniques. However, HPH-GDGTs were identified in the Amundsen sea as a significant component of the IPL-GDGT suite (e.g. Figure 4 of original submitted manuscript). This suggests that re-analysis of SPM from the Amundsen sea would yield greater concentrations of IPL-GDGTs and potentially even higher relative abundances of HPH-GDGT relative to MH and DH.

[Figure]

Figure 1. A: Boxplot showing semi-quantitative concentrations of total IPL-GDGTs (units/L) in the Amundsen Sea and Scotia Sea (high outlier of 200 units/L not shown). B: Boxplots showing semi-quantitative concentrations (units/L) of IPL-GDGT headgroups, monohexose (MH-GDGTs), dihexose (DH-GDGT), and hexose-phosphohexose (HPH-GDGT) in the Amundsen (white) and Scotia (grey) sea. C: Boxplot showing the ratio between IPL-GDGT headgroups including, MH:DH, DH:HPH and MH:HPH in the Amundsen (white) and Scotia (grey) seas. Samples where IPL-GDGTs were below detection limit of instrument are excluded from analysis. Black circles indicate outliers, white triangles indicate mean values, black line indicates median values.

We acknowledge that there may be some differences in IPL-GDGT recovery between the Amundsen and Scotia sea samples due to differences in extraction and work-up technique. However, we propose that comparison can still be made between the two seas as we do not report absolute quantities of IPL-GDGTs as the methods are semi-quantitative, we do not report the occurrence of cren', and GDGT-3 was below the detection limit of the instrument. The exceptionally high relative abundance of HPH-GDGTs in the Amundsen Sea suggests that our study does characterise the IPL-GDGT suite. Furthermore, our observations of active IPL-GDGT synthesis throughout the water column in the Amundsen sea are still relevant despite these biases in sample work-up techniques.
We have summarised this information from lines 239-252.

*2. It is important to understand the main driver of Southern Ocean GDGT distributions. However, there is a lack of strong relative discussion. For example, it is unclear how circumpolar deep water affects IPL distributions in Amundsen Sea (line 398-469). An improvement in this aspect would be helpful.*
We acknowledge that some of the significance of this study is not discussed, therefore, we have restructured the discussion, separating out the surface trends from the CDW trends (section 4.3, lines 470-524). Furthermore, we have expanded the discussion of the CDW trends to take into account the reviewers concern (lines 608-623).

*3. The authors are suggested to specify the main results in the Abstract. How did IPL-GDGT signatures shift in the study areas? And how did the signatures correlate with physicochemical parameters?*
We have updated the abstract to reflect more details of the results of the GDGT distributions including the high relative abundance of hydroxylated GDGTs, low cyclic diversity and the implications of active GDGT synthesis in the CDW (lines 28-39).

*4. The structure of some sections should be revised. Please move the section 1.2 to Methodology and remove the subtitle of 1.1. Also, the Result section should be condensed by, for example, removing redundant sentences, e.g., repeating results on water masses (line 236 and 241).*

**We have edited manuscript according to reviewer 1's recommendations. Section 1.2 (study area) has been moved to the methods (lines 160-202). We've kept the introduction subtitle (1.1 Tracing Archaea with Intact Polar Lipids) as the introduction becomes long without it. We have condensed the results section by removing section 3.2 (Intact Polar GDGT inventory) and general editing throughout.**

*5. The conclusion is suggested to be re-written to sufficiently show the significance of this study.*

**We have edited the conclusions to highlight the main findings of our paper (lines 676-692).**

*Some detailed suggestions are given below:*
*Line 109 and line 233: A colon should be used instead of semicolon.*
***This has been corrected***

*Line 110: should it be SFACC?*

**SAACF is a widely used abbreviation for the Southern Front of the ACC both within the Antarctic research field and within the Biogeosciences Journal.**

*Line 162-166: it is possible to quantify the proportion of 0.2-0.7 _m microbes?*

**Unfortunately it is not possible to quantify the number of microbes collected on the filters at the time of sampling.**

*Line 222-223: please check the use of semicolon: —- are not available, therefore–.*
**This has been corrected**

*Line 238: please check the unit of oxygen concentration.*

**We have corrected this typo to µmol/Kg. we have also converted the dissolved oxygen concentration units of the Amundsen sea data set from ml/L to µmol/Kg for consistency between the data sets.**

*Line 283: Why to present the results of MH/DH and MH/HPH ratios? It is not introduced in the Introduction or discussed later.*

**We present these data as ratios as an additional way to explore the data. As MH is predominantly a degradation product, we thought it could be interesting to see how the ratio of this headgroup changes with the more labile headgroups. However, as these do not form part of the main discussion in the paper we have removed reference to them to improve readability. As a result of this edit we have removed tables 5 and 6, and added the GDGT-0/cren and ring index ratios as additional columns to tables 1 and 2 (lines 1156-1179).**

*L288-293 and line 308-319: Please clarify to which figure or table these results refer.*
**Lines 288-293 reference table 3 and Fig. 3 (now table 4 and fig 4b)**
**Lines 308-319 reference Fig. 5 and Supplement C.**
**We have edited supplement C to include the correlation coefficients (table 2 of supplement C).**
**We have edited the text to reflect this.**

*L293-295: This statement is inconsistent with the information in Fig. 3b, which shows that the relative abundance of the HPH head group decreased from CTD station no. 1*

*to no. 3.*
**The MH and HPH in the legend of Figure 3b had been mislabelled. We have corrected the legend and therefore, the results are now consistent. The corrected figure is now figure 4b in the tracked changed manuscript.**

*Line 302: Are samples 1 and 25 equal to CTD 1 and 25?*
**Yes: this has been clarified in text**

*Line 332-333: this information is already presented in the last paragraph.*
**This section has been edited for clarity**

*Line 349-350: repeated information, please revise this sentence.*
**This section has been edited for clarity**

*Line 411: – have shown an increase in group I ——*
**This section has been edited for clarity**

*Line 413-418: are there any data of bacterial or algal community, and environmental factors from the same cruise? These data of biotic and abiotic factors would strongly support the discussion here.*
**We agree that bacterial/algal community data would improve the discussion. However, this data was not collected (beyond the CTD measurements) for either the Scotia Sea or Amundsen Sea cruises, since these expeditions were focussed on palaeoceanography and ice sheet history, respectively.**

*Line 429: please specify what use of this head group as a biomarker for the archaeal community.*
**Our statement referred to DH-cren being a potential marker for the activity of the archaeal community (i.e. high DH-cren could mean an abundance of ammonia oxidising archaea) rather than as a biomarker *proxy* for an environmental variable. We have restructured some of the text to bring the content of these sentences (Line 502-505).**

*Line 430-431: Is this conclusion made based on the results from previous studies stated on line 423-425? If yes, how to exclude the possibility of the MH head group synthesized recently by archaea?*
**The conclusion in lines 430-431 is based on the previous work by Lengger et al. (2013, 2014). At PS104/007 we only observe the monohexose (MH) head group. Previously analysed IPL-GDGT profiles of marine mesophiles (e.g. Elling et al., 2017; Sinninghe Damsté et al., 2012; Bale et al., 2019) report MH-GDGT co-occurring with HPH and DH (Elling et al., 2017; Sinninghe Damsté et al., 2012; Bale et al., 2019). Therefore, if the MH-GDGTs detected at PS104/007 were from a recently active archaeal population then we would expect to also observe other IPL-GDGTs (e.g. HPH-GDGTs). The results from the surface of PS104/007 are in contrast with PS104/003 and Scotia sea CTD 1 where MH co-occurs with DH and HPH, suggesting active archaeal populations in the surface water depths.**

*Line 438-442: are these data of productivity, nutrient concentration and light intensity from the same cruise with the present study?*
**No nutrient data is available for the cruise where our samples were collected. We have edited the text to clarify that these data referenced are not from the same cruise (line 514).**

*Line 447-448: it is not clear how figure 4 shows an increase in diversity downwards in IPL-GDGT. As I see it, the diversity is also high in the surface layer as shown by bars.*

**We acknowledge that figure does not demonstrate our point that IPL-GDGT diversity increases with depth as we only show the head group. This point is better demonstrated by Table 4 and therefore we have updated the text to reflect this (line 585, table relabelled as 3).**

*Line 455 and line 498: what does the word 'This' exactly mean? Please clarify.*
**We have edited and clarified.**

*Line 466-467: the difference between stations?*
**This sentence has been edited and clarified to indicate that we are discussing differences in depth.**

*Line 468: It is not clear what the small contrast in HPH and DH-cren distribution is.*
**This sentence has been edited and clarified with 603-605.**

*Line 471-474: These sentences should be moved to the Method or Result section.*
**This has been corrected**

*Line 516: — was detected —*
**This has been corrected**

*Fig. 1: This figure should be improved. For example, the color and size of the names of ocean fronts should be changed. Fig. 2: Consider to use different colors for different water masses and write the Amundsen Sea and the Scotia Sea in the panel. Fig. 3: It is not clear why GDGT concentration is presented as units/L.*
**We have edited these figures following the reviewer's suggestions.**

**Figure 1 has been re-coloured to improve clarity.**

**Figure 2 water masses have been shaded based on ocean temperature.**

**Figure 3 (now relabelled as 4): GDGT concentration is presented at units/L as the method is semiquantitative (as mentioned in lines 222-224), despite this concentration can be compared between samples but should not be considered absolute. Therefore, we omit the units to prevent misleading the reader. This approach has been used in other IPL-GDGT publications (e.g. Besseling et al., 2019).**

References Used in Discussion

Bale, N.J., Palatinszky, M., Rijpstra, W.I.C., Herbold, C.W., Wagner, M. & Sinninghe Damsté, J.S.: Membrane lipid composition of the moderately thermophilic ammonia-oxidizing archaeon "Candidatus Nitrosotenuis uzonensis" at different growth temperatures. 85, 1-17, http:// doi.org/10.1128/AEM.01332-19, 2019.

Besseling, M.A., Hopmans, E.C., Koenen, M., van der Meer, M.T.J., Vreugdenhil, S., Schouten, S., Sinninghe Damsté, J.S. & Villanueva, L.: Depth-related differences in archaeal populations impact the isoprenoid tetraether lipid composition of the Mediterranean Sea water column. Org. Geochem. 135, 16-31, http://doi.org/10.1016/j.orggeochem.2019.06.008. 2019.

Elling, F.J., Konneck, M., Nicol, G.W., Stieglmeier, M., Bayer, B., Spieck, E., de la Torre, J.R., Becker, K.W., Thomm, M., Prosser, J.I., Herndl, G.J., Schleper, C. & Hinrichs, K.U.: Chemotaxonomic characterisation of the thaumarchaeal lipidome. Environ. Microbiol. 19, 2681-2700, http://doi.org/10.1111/1462-2920.13759, 2017.

Kalanetra, K.M., Bano, H. & Hollibaugh, J.T.: Ammonia-oxidizing Archaea in the Arctic Ocean and Antarctic coastal waters. Environ. Microbiol. 11, 2434-2445, http://doi.org/10.1111/j.1462-2920.2009.01974.x, 2009.

Kim, J.H., Crosta, X., Willmott, V., Renssen, H., Bonnin, J., Helmke, P., Schouten, S. & Sinninghe Damsté.: Holocene subsurface temperature variability in the eastern Antarctic continental margin. Geophys. Res. Lett. 39, 1-6, http:// doi.org/10.1029/2012gl051157, 2012.

Lengger, S.K., Hopmans, E.C., Sinninhe Damsté, J.S. & Schouten, S.: Comparison of extraction and work up techniques for analysis of core and intact polar tetraether lipids from sedimentary environments. Org. Geochem. 47, 34-40, http://doi.org/10.1016/j.orggeochem.2012.02.009, 2012.

Lennger, S.K., Kraaij,M., Tjallingii, R., Baas, M., Stuut, J.B., Hopmans, E.C., Sinninghe Damsté, J.S. & Schouten, S.: Differential degradation of intact polar and core glycerol dialkyl glycerol tetraether lipids upon post-depositional oxidation. Org. Geochem. 65, 83-93, https://doi.org/10.1016/j.orggeochem.2013.10.004, 2013.

Lennger, S.K., Hopmans, E.C., Sinninghe Damsté, J.S. & Schouten, S.: Fossilization and degradation of archaeal intact polar tetraether lipids in deeply buried marine sediments (Peru Margin). Geobiology. 12, 212-220, http:// doi.org/10.1111/gbi.12081, 2014.

Schouten, S., Hopmans, E.C., Rosell-Melé, A., Pearson, A., Adam, P., Bauersachs, T., Bard, E., Bernasconi, S.M., Bianchi, T.S., Brocks, J.J., Carlson, L.T., Castañeda, I.S., Derenne, S., Selver, A.D., Dutta, K., Eglinton, T., Fosse, C., Galy, V., Grice, K., Hinrichs, K.U., Huang, Y., Huguet, A., Huguet, C., Hurley, S., Ingalls, A., Jia, G., Keely, B., Knappy, C., Kondo, M., Krishnan, S., Lincoln, S., Lipp, J., Mangelsdorf, K., Martínez-García, A., Ménot, G., Mets, A., Mollenhauer, G., Ohkouchi, N., Ossebaar, J., Pagani, M., Pancost, R.D., Pearson, E.J., Peterse, F., Reichart, G.J., Schaeffer, P., Schmitt, G., Schwark, L., Shah, S.R., Smith, R.W., Smittenberg, R.H., Summons, R.E., Takano, Y., Talbot, H.M., Taylor, K.W.R., Tarozo, R., Uchida, M., Van Dongen, B.E., Van Mooy, B.A.S., Wang, J., Warren, C., Weijers, J.W.H., Werne, J.P., Woltering, M., Xie, S., Yamamoto, M., Yang, H., Zhang, C.L., Zhang, Y., Zhao, M. & Sinninghe Damsté, J.S.: An interlaboratory study of TEX86 and BIT analysis of sediments, extracts, and standard mixtures, Geochem. Geophys. Geosyst., 14, 5263-5265, https://doi.org/10.1002/2013GC004904, 2013.

Sinninghe Damsté, J.S., Rijpstra, W.I.C., Hopmans, E.C., Jung, M.Y., Kim, J.G., Rhee, S.K., Stieglmeier, M. & Schleper, C.: Intact Polar and Core Glycerol Dibiphytanyl Glycerol Tetraether Lipids of Group I.1a and I.1b Thaumarchaeota in Soil. 78, 6866-6874, http://doi.org/10.1128/aem.01681-12, 2012.

Weber, Y., Sinninghe Damsté, J.S., Hopmans, E.C., Lehmann, M.F. & Niemann, H.: Incomplete recovery of intact polar glycerol dialkyl glycerol tetraethers from lacustrine suspended biomass. Limn. Oceanogr. Methods. 15, 782-793. http://doi.org/10.1002/lom3.10198

**BG-2020-333** *Archaeal Intact Polar Lipids in Polar Waters: A Comparison Between the Amundsen and Scotia Seas*

**Reply to Reviewer 2**

On behalf of the co-authors, I thank this anonymous referee for their helpful comments and the time they took to improve the paper. We reply to the reviewer's comments in bold below.

*This study attempts to assess the distribution of core GDGTs and intact polar lipid (IPL)-GDGTs in the waters of the Scotia Sea and Amundsen Sea in the Southern Ocean. GDGTs are membrane spanning isoprenoidal lipids that make up a significant portion of the membrane bilayer of a variety of Archaea. Modifications to these core membrane lipids, including the addition of 1-8 cyclopentane rings, 1 cyclohexane ring, or hydroxylations, or changes to the polar head groups, are thought to represent physiological responses to environmental factors, such as changes in temperature, pH, or redox. Because of this physiological connection between GDGT modifications and environmental conditions, an because GDGTs can be well preserved in ancient sediments, these molecules have been employed as paleotemperature proxies. In addition, some of these modified GDGTs have also been proposed to be restricted to certain archaeal groups and, thus, have been utilized as diagnostic markers for specific archaeal group in a environmental settings. However, the utility of GDGTs and IPL-GDTGs as proxies and/or diagnostic markers requires an understanding of a variety of factors – confirming the correlation between environmental factors and the specific modifications made on the GDGT structures, determining the distribution of various GDGT structures in different ecosystems, and assessing the occurrence of specific GDGT structures in different cultured archaeal groups.*

*In this study, the authors investigate the occurrence of the core GDGT structures, which are most relevant for paleotemperature proxies, as well as the occurrence of the GDGT structures with various polar headgroups in the Scotia Sea and Amundsen Sea.*
**We wish to clarify that in our study we only analyse the GDGTs as IPLs. We explicitly mention that we did not measure GDGTs as their core components in lines 287.**

*The analyses performed in this study are well done and provide an interesting picture of the distribution of GDGTs in the Southern Ocean. In particular, they show limited cyclization of GDGTS in their samples with the majority of core GDGTs having zero rings. In addition, they see a significant amount of hydroxylated GDGTs which have been proposed to function in helping maintain membrane fluidity at low temperatures. The authors infer that both observations may reflect the cold environment of the Southern Ocean which their specific sites can range from -1 to 8 degrees Celsius. The occurrence of IPLs is a little more difficult to parse. Although I agree that IPLs can represent the occurrence of living archaea in the water column, I am not convinced that the intact IPLs are useful as diagnostic markers specifically for the Thaumarchaeota as I believe other archaea are known to produce head groups with 1 or 2 hexose groups.*
**Our interpretation of (recently) living Thaumarchaeota in the water column is based on both the structure of the core GDGT lipid and the polar head group. Crenarchaeol has been established as a biomarker for Thaumarchaeota, having been identified in a large number of pure cultures (see Schouten et al., 2013 for review). Intact polar lipids with monohexose, dihexose, and hexose phosphohexose head groups associated with a crenarchaeol core GDGT have also been identified in Thaumarchaeota pure cultures (e.g. Elling et al., 2017; Bale et al., 2019). We interpret the occurrence of (recently) living Thaumarchaeota based on the combination of IPL head group with crenarchaeol core lipid.**

*Nonetheless, the authors are able to demonstrate some interesting IPL-GDGT patterns that may reflect temporal changes in archaeal communities. For example, in the surface samples*

*form the Amundsen Sea (collected within the euphotic zone), there was an absence of IPL-GDGTs. Previous studies have shown the absence of archaea in the surface waters of the Southern Ocean (and large abundance of bacteria) and this lack of IPL-GDGTs corresponded well with that larger seasonal variation in archaeal populations. Overall, this study is well-designed and well-written and contributes some significant knowledge into the environmental distribution of both core GDGTs and IPLGDGTs.*

References Used in Discussion

Bale, N.J., Palatinszky, M., Rijpstra, W.I.C., Herbold, C.W., Wagner, M. & Sinninghe Damsté, J.S.: Membrane lipid composition of the moderately thermophilic ammonia-oxidizing archaeon "Candidatus Nitrosotenuis uzonensis" at different growth temperatures. 85, 1-17, http://10.1128/AEM.01332-19, 2019.

Elling, F.J., Konneck, M., Nicol, G.W., Stieglmeier, M., Bayer, B., Spieck, E., de la Torre, J.R., Becker, K.W., Thomm, M., Prosser, J.I., Herndl, G.J., Schleper, C. & Hinrichs, K.U.: Chemotaxonomic characterisation of the thaumarchaeal lipidome. Environ. Microbiol. 19, 2681-2700, http://10.1111/1462-2920.13759. 2017.

Schouten, S., Hopmans, E.C. & Sinninghe Damste, J.S.: The organic geochemistry of glycerol dialkyl glycerol tetraether lipids: A review. Org. Geochem. 54, 19-61, http://10.1016/j.orggeochem.2012.09.006, 2013.